# WHEN SELECTION MEETS INTERVENTION: ADDITIONAL COMPLEXITIES IN CAUSAL DISCOVERY

**Haoyue Dai**[1,2]    **Ignavier Ng**[1]    **Jianle Sun**[1]    **Zeyu Tang**[1]

**Gongxu Luo**[2]    **Xinshuai Dong**[1]    **Peter Spirtes**[1]    **Kun Zhang**[1,2]

[1]Carnegie Mellon University    [2]Mohamed bin Zayed University of Artificial Intelligence

## ABSTRACT

We address the common yet often-overlooked selection bias in interventional studies, where subjects are selectively enrolled into experiments. For instance, participants in a drug trial are usually patients of the relevant disease; A/B tests on mobile applications target existing users only, and gene perturbation studies typically focus on specific cell types, such as cancer cells. Ignoring this bias leads to incorrect causal discovery results. Even when recognized, the existing paradigm for interventional causal discovery still fails to address it. This is because subtle differences in *when* and *where* interventions happen can lead to significantly different statistical patterns. We capture this dynamic by introducing a graphical model that explicitly accounts for both the observed world (where interventions are applied) and the counterfactual world (where selection occurs while interventions have not been applied). We characterize the Markov property of the model, and propose a provably sound algorithm to identify causal relations as well as selection mechanisms up to the equivalence class, from data with soft interventions and unknown targets. Through synthetic and real-world experiments, we demonstrate that our algorithm effectively identifies true causal relations despite the presence of selection bias.

## 1    INTRODUCTION

Experimentation is often seen as the gold standard for discovering causal relations, but due to its cost, alternative methods have been developed to infer causality from pure observational data (Spirtes et al., 2000; Pearl, 2009). Many real-world scenarios fall between these two extremes, involving passive observations collected from interventions. Interventional causal discovery addresses this, with methods for *hard* interventions (Cooper & Yoo, 1999; Hauser & Bühlmann, 2015), *soft* interventions (Tian & Pearl, 2001; Eberhardt & Scheines, 2007), and those with unknown targets (Eaton & Murphy, 2007; Squires et al., 2020). A detailed review is provided in §2.1, with further related works in Appendix D.

While significant progress has been made in interventional causal discovery, existing methods overlook a critical issue: *selection bias* (Heckman, 1977; 1990; Winship & Mare, 1992). Though ideally, experiments should be randomly assigned in the general population, in practice, subjects are usually *pre-selected*. For example, drug trial participants are typically patients with the relevant disease; A/B tests target only existing users, and gene perturbation studies often focus on specific cell types like cancer cells. Ignoring this bias leads to incorrect statistical inferences. While various methods address selection bias for causal inference (Didelez et al., 2010; Bareinboim & Pearl, 2012; Bareinboim et al., 2014) and for observational causal discovery (Spirtes et al., 1995; Borboudakis & Tsamardinos, 2015; Zhang et al., 2016), no existing work tackles selection bias in interventional causal discovery.

Then, one may naturally wonder whether existing well-established paradigms from both interventional causal discovery and observational causal discovery with selection bias can solve the problem. However, as we will illustrate in Examples 1 and 2, these methods still fail to characterize data given by intervention under selection, and will thus lead to false causal discovery results. This is because subtle differences in *when* and *where* interventions happen can lead to significantly different statistical patterns, demanding a new problem setup and model. This is exactly what we address in this paper.

**Contributions:** We introduce a new problem setup for interventional causal discovery with selection bias. We show that existing graphical representation paradigms fail to model data under selection, since the *when* and *where* of interventions have to be explicitly considered (§2). To this end, we propose a new graphical model that captures the dynamics of intervention and selection, characterize its Markov properties, and provide a graphical criterion for Markov equivalence (§3). We develop a

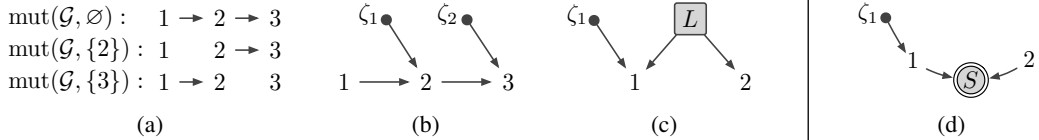

Figure 1: Examples of the existing graph representations. (a) and (b) show *mutilated DAGs* (Hauser & Bühlmann, 2012) and the *augmented DAG* (Yang et al., 2018) for $\mathcal{G} = 1 \to 2 \to 3$ with targets $\mathcal{I} = \{\varnothing, \{2\}, \{3\}\}$. Solid nodes represent the intervention indicators. (c) is the augmented DAG for $\mathcal{G} = 1 \leftarrow L \to 2$ where $L$, in a square, is latent, and $\mathcal{I} = \{\varnothing, \{1\}\}$ (Magliacane et al., 2016). (d) shows a seemingly natural representation for selection bias, $\mathcal{G} = 1 \to S \leftarrow 2$ where $S$, in double circles, is selected, and $\mathcal{I} = \{\varnothing, \{1\}\}$. But does (d) truly capture the underlying process? See Example 1.

sound algorithm to identify causal relations and selection mechanisms up to the equivalence class, from data with soft interventions and unknown targets (§4). We demonstrate the effectiveness of our algorithm using synthetic and real-world datasets on biology and education (§5).

## 2 MOTIVATION

In this section, we first revisit the established paradigm graph representations for interventional causal discovery (§2.1), then use illustrative examples to demonstrate how directly extending this paradigm fails to address the selection bias issue (§2.2), followed by an analysis of why this occurs (§2.3).

### 2.1 REVISITING THE ESTABLISHED PARADIGM OF INTERVENTIONAL CAUSAL DISCOVERY

We start with the problem setup. Let the DAG $\mathcal{G}$ on vertices $[D] := \{1, \cdots, D\}$ represent a causal model where vertices correspond to random variables $X = (X_i)_{i=1}^D$. For any subset $A \subset [D]$, let $X_A := (X_i)_{i \in A}$ and by convention $X_\varnothing \equiv 0$. Interventional causal discovery aims to learn the structure of $\mathcal{G}$ from data collected under multiple intervention settings, each with an *intervention target* $I \subset [D]$, meaning variables $X_I$ are intervened on. Let $\mathcal{I} = \{I^{(0)}, I^{(1)}, \ldots, I^{(K)}\}$ denote the collection of intervention targets, and $\{p^{(0)}, p^{(1)}, \ldots, p^{(K)}\}$ the corresponding *interventional distributions* over $X$. We assume throughout $I^{(0)} = \varnothing$, i.e., the pure observational data is available.

For hard interventions, Hauser & Bühlmann (2012) consider each $p^{(k)}$ as factoring to a *mutilated DAG* over $[D]$, denoted by $\mathrm{mut}(\mathcal{G}, I^{(k)})$, where edges incoming to target $I^{(k)}$ in $\mathcal{G}$ are removed and other edges remain, as in Figure 1a. They show that two DAGs are Markov equivalent under $\mathcal{I}$ if and only if $\forall k = 0, \cdots, K$, their corresponding mutilated DAGs have the same skeleton and v-structures.

For soft interventions, however, mutilated DAG representation fails, as interventions may not remove edges, and all settings may factor to a same $\mathcal{G}$. Instead of checking each setting individually, a better approach is then to compare changes and invariances across settings: intervening on a cause changes the marginal $p(\text{effect})$, but the conditional $p(\text{effect}|\text{cause})$ remains invariant. Conversely, intervening on an effect leaves $p(\text{cause})$ unchanged, while $p(\text{cause}|\text{effect})$ changes (Hoover, 1990; Tian & Pearl, 2001). Such invariance is exploited in the *invariance causal inference framework* (Rothenhäusler et al., 2015; Meinshausen et al., 2016; Ghassami et al., 2017), typically as parametric regression analysis.

Invariance can also be understood nonparametrically. To model "the action of changing targets", Newey & Powell (2003); Korb et al. (2004) introduce the *augmented DAG*, denoted by $\mathrm{aug}(\mathcal{G}, \mathcal{I})$, which, as shown in Figure 1b, extends the original $\mathcal{G}$ by adding exogenous vertices $\zeta = \{\zeta_k\}_{k=1}^K$ as *intervention indicators*, each pointing to its target $I^{(k)}$. Whether the $k$-th intervention alters a conditional density $p(X_A|X_C)$ is then nonparametrically represented by the conditional independence (CI) relation $\zeta_k \perp X_A|X_C$ in the pooled data of $p^{(0)}$ and $p^{(k)}$, and graphically by the d-separation $\zeta_k \perp_d A|C, \zeta_{[K]\setminus\{k\}}$ in $\mathrm{aug}(\mathcal{G}, \mathcal{I})$. Yang et al. (2018) show that two DAGs are Markov equivalent under soft interventions $\mathcal{I}$ if and only if their augmented DAGs have the same skeleton and v-structures.

Such invariance analysis, together with the augmented DAG representation, offers a unified way to understand interventional data or, more generally, data from multiple domains with changing mechanisms. For example, unknown target is no longer a challenge; $\mathcal{I}$ (i.e., where changes occur) can be learned by discovering the adjacencies between $\zeta$ and $X$ (Zhang et al., 2015; Huang et al., 2020; Mooij et al., 2020; Squires et al., 2020). Hidden confounders can also be incorporated by introducing latent variables into augmented DAGs, as shown in Figure 1c. Algorithms like FCI (Spirtes et al., 2000; Zhang, 2008) are then used to for discovery (Magliacane et al., 2016; Kocaoglu et al., 2019). Despite different specifics, these problems share a same core concept under the augmented graph paradigm.

Now finally, let us consider the issue of selection bias. Although, to our knowledge, no prior work has addressed it, a seemingly natural solution is to follow the paradigm and introduce selection variables into augmented DAGs, as shown in Figure 1d. This seems intuitive, especially given Figure 1c, as the FCI algorithm can indeed handle both hidden confounders and selection bias. However, does this augmented graph truly capture the underlying dynamics? The answer is no – or at least, it is not straightforward. Let us examine the following Examples 1 and 2 that illustrate these complexities.

## 2.2 How the Augmented DAG Paradigm Fails in the Presence of Selection

We start with an example of a clinical study where only patients with a specific disease are involved.

**Example 1.** Consider a clinical study focusing on two variables: $X_1$ (Blood Glucose Levels) and $X_2$ (Hearing Ability). Unknown to the doctor, these variables are independent with no causal relations or confounders. This independence is shown in the scatterplot (both '$\times$' and '$\bullet$') in (a) of Figure 2 (hereafter omitted), where $X_1$ and $X_2$ are independently drawn from $\cup[1, 2]$. However, the study somehow only includes Alzheimer's

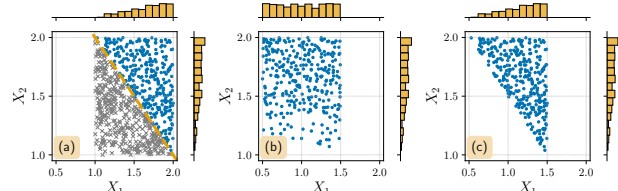

Figure 2: (a) Scatterplot of $X_1$; $X_2$ in general population (both '$\times$' and '$\bullet$'), with only '$\bullet$' individuals involved into study as $p^{(0)}$. (b) and (c) show $p^{(1)}$ after two distinct but both effective interventions on $X_1$, applied to '$\bullet$' from (a).

Disease (AD) patients. Since both variables contribute to AD, only individuals with a high combined value of the two (say, $X_1 + X_2 \geq 3$) were included in the study, represented by the '$\bullet$'s in (a).

In the trial, patients are randomly assigned to either a placebo or a treatment controlling Blood Glucose Levels, i.e., $\mathcal{I} = \{\varnothing, \{1\}\}$. The control group $p^{(0)}$ follows the '$\bullet$'s distribution in (a). To model the intervention, we can use a hard stochastic one, randomly assigning $X_1$ a lower value, shifting each '$\bullet$' $(X_1, X_2)$ in (a), i.e., individuals before their treatments, to $(X_1', X_2)$ in (b), with $X_1' \sim \cup[0.5, 1.5]$ and importantly, $X_2$ remains unchanged, as indeed $X_1$ does not influence $X_2$. Alternatively, a 'soft' intervention can be modeled, reducing $X_1$ relative to its current value, e.g., to $(X_1 - 0.5, X_2)$, resulting in scatterplots (c). The interventional distribution $p^{(1)}$ follows (b) or (c).

Then, can we directly apply the augmented graph in Figure 1d, since it is exactly independent $X_1$ and $X_2$ selected to the trial, and $X_1$ is intervened on? According to it, two statements should hold:

1. The marginal $p(X_2)$ should change by the intervention, since $\zeta_1 \not\perp_d 2 | S$;
2. The conditional $p(X_2|X_1)$ should remain invariant, since $\zeta_1 \perp_d 2|1, S$;

However, our observations contradict both. Comparing (a) with (b) or (c), $p(X_2)$ remains unchanged, as seen in the marginal density plots, while $p(X_2|X_1)$ changes. For example, at $X_1 = 1.25$, $X_2$ in (a) follows $\cup[1.75, 2]$, in (b) is non-uniform, more concentrated at higher values, and in (c) follows $\cup[1.5, 2]$. This discrepancy suggests that the graph in Figure 1d does not fit the data here. One augmented graph however indeed fits is $\zeta_1 \to 1 \leftarrow 2$, meaning that if directly applying existing standard algorithms, the doctor would falsely conclude that Hearing Ability causes Blood Glucose Levels. △

We show that simply augmenting the DAG with selection variables fails to model interventional data under selection. In Example 1 there is at least another (incorrect) augmented DAG that fits the data. In contrast, in Example 2, no augmented DAG fits the data, suggesting a failure of this paradigm.

**Example 2.** Let $X_1$, $X_2$, and $X_3$ denote the number of bird-nesting shrubs, predatory birds, and pests, respectively. An ecologist is studying pest issues in several fields across the state. After collecting data, denoted $p^{(0)}$, the ecologist finds that $X_1$ and $X_3$ are conditionally independent given $X_2$. This aligns with the true causal relations $1 \to 2 \to 3$, where shrubs attract birds, birds reduce pests, but shrubs do not directly affect pests. To control pests, the ecologist plants more shrubs (intervening on $X_1$) in these fields and, after some time collect data, denoted $p^{(1)}$. As expected, there is an increase in $X_2$ and decrease in $X_3$. However, further analysis reveals a surprise: conditioning on $X_2$, $X_1$ and $X_3$ now appear dependent in $p^{(1)}$, with more shrubs associated with less pests. This confuses the ecologist – "did I find a new type of shrubs that directly reduce pests, i.e., added a direct edge $1 \to 3$?"

"Selection may be at play!", suggested a student, noting the study focused only on fields with pest issues, i.e., high $X_3$ values. The initial $p^{(0)}$ follows a DAG $1 \to 2 \to 3 \to S$ where d-separation $1 \perp_d 3|2, S$ indeed holds, consistent with the observed CI in $p^{(0)}$. But after intervening on $X_1$, shouldn't

$p^{(1)}$ still be Markovian to this DAG? Why did the CI disappear? No augmented graph can explain this anomaly, as it suggests that each $p^{(k)}$, conditioned on root variables $\zeta$, should still be Markovian to the original DAG. This puzzle is unsolved until explained in Example 3: there is no specialty with the shrubs. A simulation on this example, similar to Figure 2 above, is provided in Appendix C.1. △

## 2.3 WHY THE AUGMENTED DAG PARADIGM FAILS WHEN SELECTION PRESENTS

The core reason why augmented DAG paradigm fails, as illustrated in §2.2, lies in the timing and context – *when* and *where* interventions are applied. In real-world scenarios, interventions are usually applied *after* selection, as experiments are typically designed for specific scopes of study. When selection is interpreted as survival, this means an individual must first survive itself, *before* undergoing any observations or interventions (e.g., '•'s in Figure 2a). We consider this setting in our work.

Simply extending the augmented graph with selection variables (as in Figure 1d), however, models a different scenario: one where interventions are applied *from scratch*. There, individuals are not selected when they receive interventions (and non-interventions), and then undergo the *same selection mechanisms* afterwards, until observed. This scenario is much rarer, with examples like social programs applied to newborns and observed later in life, or medical trials on newly generated stem cells.

This fundamental distinction in *when* and *where* interventions occur is often overlooked. This is because when selection bias is absent, and even with latent variables, these two forms produce the same interventional data at the distribution level. However, now with selection it is different; the selective inclusion of individuals and their pre-intervention world must be carefully modeled.

## 3 CAUSAL MODEL INVOLVING SELECTION AND INTERVENTION

In §2 we demonstrated that the *when* and *where* of interventions matter. Building on this motivation, in this section, we define the causal graph on how interventions are applied under selection (§3.1), characterize the Markov properties (§3.2), and provide the criteria for determining whether two DAGs, possibly under selection, are Markov equivalent given possibly different interventions (§3.3).

We follow the notation in §2.1, with the key difference being that the DAG $\mathcal{G}$ is now over vertices $[D] \cup S$, where $S = (S_i)_{i=1}^T$ represents unobserved selection variables conditioned upon their specific values. W.l.o.g. let each $S_i$ be binary, has no children, and has parents only from $[D]$. Let $\mathbf{1}$ be the vector of all 1s. A sample is observed if and only if it satisfies all selection criteria, denoted by $S = \mathbf{1}$.

In the DAG $\mathcal{G}$, for any vertices $i, j \in [D] \cup S$, $i$ is a *parent* of $j$ and $j$ is a *child* of $i$ if $i \to j \in \mathcal{G}$, denoted by $i \in \mathrm{pa}_{\mathcal{G}}(j)$ and $j \in \mathrm{ch}_{\mathcal{G}}(i)$; $i$ is an *ancestor* of $j$ and $j$ is a *descendant* of $i$ if $i = j$ or there is a directed path $i \to \cdots \to j$ in $\mathcal{G}$, denoted by $i \in \mathrm{an}_G(j)$ and $j \in \mathrm{de}_G(i)$. These notations extend to sets: e.g., for any vertex set $C \subset [D] \cup S$, $\mathrm{pa}_{\mathcal{G}}(C) := \bigcup_{i \in C} \mathrm{pa}_{\mathcal{G}}(i)$. For any vertex $i \in [D]$, we say $i$ is *directly selected* if $i \in \mathrm{pa}_{\mathcal{G}}(S)$, and *ancestrally selected* if $i \in \mathrm{an}_{\mathcal{G}}(S)$.

### 3.1 CAUSAL GRAPHICAL MODEL FOR INTERVENTIONS UNDER SELECTION

Building on the principle that enrolled individuals are already selected before interventions, we introduce the following graphical model, with examples and explanations given afterwards.

**Definition 1 (Interventional twin graph).** For a DAG $\mathcal{G}$ over $[D] \cup S$ and a intervention target $I \subset [D]$, the *interventional twin graph* $\mathcal{G}^{(I)}$ is a DAG with vertices $\{\zeta\} \cup X \cup X_{\mathrm{aff}}^* \cup \mathcal{E}_{\mathrm{aff}} \cup S^*$, where[1]:

- $\zeta$ is an exogenous binary indicator for whether a sample is intervened ($\zeta = 1$) or not ($\zeta = 0$);
- $X = \{X_i\}_{i=1}^D$ are variables in the observed *reality world*, pure observational or interventional;
- $X_{\mathrm{aff}}^* = \{X_i^* : i \in \mathrm{de}_{\mathcal{G}}(I)\}_{i=1}^D$ are variables in the unobserved *counterfactual basal world*[2], representing the variable values *before* the intervention. As indicated in its name, only variables *affected* by the intervention, i.e., those in $\mathrm{de}_{\mathcal{G}}(I)$, are split into these additional vertices; unaffected variables retain identical values in both worlds and can be represented solely by $X$;
- $\mathcal{E}_{\mathrm{aff}} = \{\epsilon_i : i \in \mathrm{de}_{\mathcal{G}}(I)\}_{i=1}^D$ are common exogenous noise terms shared by the two worlds;
- $S^* = \{S_i^*\}_{i=1}^T$ represent the selection status before the intervention in the counterfactual world.

$\mathcal{G}^{(I)}$ consists of the following four types of direct edges:

- Direct causal effect edges in both worlds: for each $i \to j \in \mathcal{G}$ with $i, j \in [D]$, add $X_i \to X_j$ to $\mathcal{G}^{(I)}$. Additionally, if $i \in \mathrm{de}_{\mathcal{G}}(I)$, add $X_i^* \to X_j^*$; otherwise, if $j \in \mathrm{de}_{\mathcal{G}}(I^{(k)})$, add $X_i \to X_j^*$;

---

[1]The word "twin" is to echo the *twin network* from (Balke & Pearl, 1994). See discussions in Appendix D.
[2]The word "basal" is borrowed from biology, referring to a natural state of cells prior to any perturbations.

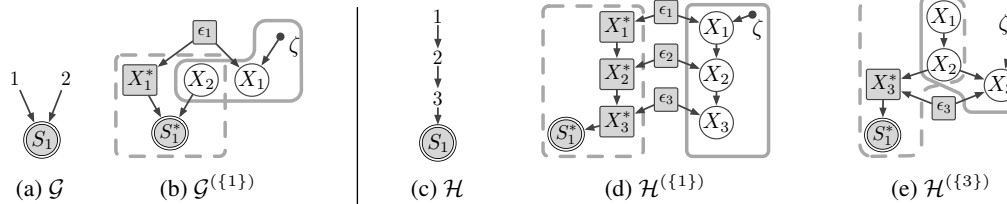

(a) $\mathcal{G}$     (b) $\mathcal{G}^{(\{1\})}$     (c) $\mathcal{H}$     (d) $\mathcal{H}^{(\{1\})}$     (e) $\mathcal{H}^{(\{3\})}$

Figure 3: Examples of interventional twin graphs (Definition 1). (a) and (c) are two DAGs for clinical and pest control Examples 1 and 2, respectively; (b) and (d), (e) are their corresponding interventional twin graphs under different targets. The white $X$ nodes and solid $\zeta$ node are observed, forming the reality world (enclosed by solid frames), where observations or interventions are conducted. The grey nodes are unobserved, of which squares ($X^*_{\mathrm{aff}}$ and $\mathcal{E}_{\mathrm{aff}}$) are latent variables and double circles ($S^*$) are selection variables. The counterfactual basal world is enclosed by dashed frames.

- Selection edges in the counterfactual basal world: for each $i \to S_j \in \mathcal{G}$ with $i \in [D]$, $j \in [T]$: if $i \in \deg_{\mathcal{G}}(I)$, add $X^*_i \to S^*_j$ to $\mathcal{G}^{(I)}$; otherwise, add $X_i \to S^*_j$;
- Edges representing common exogenous influences: $\{\epsilon_i \to X_i, \epsilon_i \to X^*_i\}_{i \in [D] \cap \deg_{\mathcal{G}}(I)}$;
- Edges representing mechanism changes due to the intervention: $\{\zeta \to X_i\}_{i \in I}$.

Illustrative examples for interventional twin graphs (Definition 1) are shown in Figure 3. In what follows, we explain several key structural and statistical insights of this causal graphical model.

**What is modeled by the interventional twin graph?** In $\mathcal{G}^{(I)}$, only $X$ and $\zeta$ are observed, representing pure observational data, $p(X|\zeta = 0, S^* = \mathbf{1})$, and interventional data, $p(X|\zeta = 1, S^* = \mathbf{1})$. Crucially, $S^* = \mathbf{1}$ is conditioned on, meaning all individuals, observed or intervened, were selected at the outset. The key difference from the augmented graph (Figure 1d) is that, here $\mathcal{G}^{(I)}$ explicitly models each individual's unobserved pre-intervention values: non-affected variables retain their values[3] as $X$, while affected variables, whose values change, are modeled by extra vertices $X^*_{\mathrm{aff}}$. The pre- and post-intervention worlds share $\mathcal{E}$, common external influences like individual-specific traits. Specifically, selection is applied in the pre-intervention world, while observation is made post-intervention.

**According to the graph, what is changed by the intervention?** Individuals in pure observational and interventional data are not matched, which reflects common interventional studies (e.g., RNA sequencing is destructive, preventing measurements to a same cell both before and after a gene knockout). Instead, the two datasets are related at the distribution level: directed edges from $\zeta$ to $X_I$ indicate changes in generating mechanisms for targeted variables. For affected but not directly targeted variables $i \in \deg_{\mathcal{G}}(I) \backslash I$, as suggested by the graph, their generating functions $p(X_i|X_{\mathrm{pa}_{\mathcal{G}}(i)}, \epsilon_i)$ remain invariant for $\zeta = 0, 1$. However, unlike in augmented graph paradigm, this invariance no longer extends to observed conditional distributions $p(X_i|X_{\mathrm{pa}_{\mathcal{G}}(i)})$. For instance, in Figure 3b, intervening on $X_1$ alters not only $p(X_1)$ but also $p(X_2|X_1)$ and $p(X_3|X_2)$ (see Example 3).

### 3.2 THE MARKOV PROPERTIES

Our ultimate goal is to discover true causal relations and selection mechanisms from data. To this end, we must understand the CI implications in the data. Hence, in what follows, we define the Markov properties, showing how the interventional twin graph model serves to identify these CI implications.

To start, let us revisit Example 1 (clinical study) with the defined interventional twin graph $\mathcal{G}^{(\{1\})}$, as shown in Figure 3b. It becomes clear that there is $\zeta \perp\!\!\!\perp_d X_2|S^*$ and $\zeta \not\perp\!\!\!\perp_d X_2|X_1, S^*$, consistent with the invariant $p(X_2)$ and the changed $p(X_2|X_1)$, resolving the earlier discrepancy given by Figure 1d.

For discovery from data, two types of statistical information can help: 1) conditional (in)dependencies among variables within each interventional distribution, and 2) the (in)variances of conditional distributions across different interventions. Below, we formally define these relations implied by the model.

**Theorem 1 (CI and invariance implications).** *For interventional distributions $\{p^{(k)}(X)\}_{k \in \{0\} \cup [K]}$ generated from DAG $\mathcal{G}$ with targets $\{I^{(k)}\}_{k \in \{0\} \cup [K]}$, let $\{\mathcal{G}^{(I^{(k)})}\}_{k \in \{0\} \cup [K]}$ be the corresponding interventional twin graphs. For any disjoint $A, B, C \subset [D]$, the following two statements hold:*

1. *For any $k \in \{0\} \cup [K]$, if $X_A \perp\!\!\!\perp_d X_B|X_C, S^*, \zeta$ holds in $\mathcal{G}^{(I^{(k)})}$, then $X_A \perp\!\!\!\perp X_B|X_C$ in $p^{(k)}$.*
2. *For any $k \in [K]$, if $\zeta \perp\!\!\!\perp_d X_A|X_C, S^*$ holds in $\mathcal{G}^{(I^{(k)})}$, then $p^{(k)}(X_A|X_C) = p^{(0)}(X_A|X_C)$.*

---

[3]For reasons why not to also split unaffected variables across both worlds, see Appendix C.2.

Theorem 1 shows that both types of statistical information are implied by the graphical conditions, namely d-separations among $X \cup \{\zeta\}|S^*$ in interventional twin graphs. The two conditions characterize the CIs in each intervention, and the conditional invariances across interventions, respectively.

We now show how selection and intervention specifically lead to the first type of information, CIs in each distribution. First, selection is known to introduce spurious dependencies in pure observations:

**Lemma 1 (Additional dependencies induced by selections).** *For any DAG $\mathcal{G}$ on $[D] \cup S$ and disjoint $A, B, C \subset [D]$, if $A \perp\!\!\!\perp_d B|C, S$ holds, then $A \perp\!\!\!\perp_d B|C$ also holds. The reverse is not necessarily true.*

Further, interventions can introduce even more dependencies, making interventional distributions no longer Markovian to the original DAG $\mathcal{G}$ (recall the post-pest-control distribution in Example 2):

**Lemma 2 (Even more dependencies induced by interventions).** *For any DAG $\mathcal{G}$ on $[D] \cup S$, target $I \subset [D]$, and disjoint $A, B, C \subset [D]$, if $X_A \perp\!\!\!\perp_d X_B|X_C, S^*, \zeta$ holds in the twin graph $\mathcal{G}^{(I)}$, then $A \perp\!\!\!\perp_d B|C, S$ holds in the original DAG $\mathcal{G}$. The reverse is not necessarily true, except when $I = \varnothing$.*

Lemma 2 offers a counterintuitive insight that contrasts with cases without selection or where selection is not applied before interventions, as those modeled by the augmented graph paradigm. In those cases, interventions add no dependencies and may only add independencies (e.g., via hard interventions). In our setting, however, an intervention may indeed add more dependencies. Now, let us examine Theorem 1 and Lemma 2 in the context of Example 2 (pest control), address the ecologist's concern, and summarize the motivations behind the Markov properties in this subsection.

**Example 3.** Continuing from Example 2, the original DAG $\mathcal{H}$ and the interventional twin graph $\mathcal{H}^{(\{1\})}$ are shown in Figures 3c and 3d, respectively. To explain the $X_1 \not\perp\!\!\!\perp X_3|X_2$ observed after intervening on shrubs, graphically we indeed see the d-connection $X_1 \not\perp\!\!\!\perp_d X_3|X_2, S^*, \zeta$ in $\mathcal{H}^{(\{1\})}$, with an open path $X_1 \leftarrow \epsilon_1 \rightarrow X_1^* \rightarrow X_2^* \rightarrow X_3^* \leftarrow \epsilon_3 \rightarrow X_3$ where the collider $X_3^*$ has its descendant $S_1^*$ conditioned on. Statistically, $X_3^*$ is a combination of exogenous noises $\epsilon_1, \epsilon_2, \epsilon_3$, and selection on $X_3^*$ renders $\epsilon_1, \epsilon_2, \epsilon_3$ not independent anymore. Such dependent noises also leave trace in conditional invariances: though only $X_1$ is targeted, not only $p(X_1)$ but also $p(X_2|X_1)$ and $p(X_3|X_2)$ are altered, as graphically implied by the $\zeta \not\perp\!\!\!\perp_d X_1|S^*$, $\zeta \not\perp\!\!\!\perp_d X_2|X_1, S^*$, and $\zeta \not\perp\!\!\!\perp_d X_3|X_2, S^*$, respectively.

If, however, the ecologist somehow directly targets $X_3$ (pests), the corresponding $\mathcal{H}^{(\{3\})}$ (see Figure 3e) shows that $X_1 \perp\!\!\!\perp X_3|X_2$ holds in the interventional distribution this time, and the marginals of $X_1, X_2$ remain unaltered. More details on this pest-control example are in Appendix C.1. $\quad\triangle$

## 3.3 MARKOV EQUIVALENCE RELATIONS

In §3.2 we characterized the CI relations implied by the true model in the data. Now, to identify the true model from data, in this section, we must understand to what extent the true model is *identifiable*, as different models may share identical CI implications, namely, being *Markov equivalent* (Definition 2). To establish graphical criteria for this equivalence, we leverage the maximal ancestral graph (MAG) framework. In §3.3.1, we introduce MAG basics. In §3.3.2, we characterize the MAGs of interventional twin graphs, and accordingly, present the graphical criteria for Markov equivalence.

We first define the Markov equivalence. Two different DAGs with different intervention targets (since we allow unknown targets) can entail the same CI and invariance relations in the data. Formally,

**Definition 2 (Markov equivalence).** Let $\mathcal{G}$ and $\mathcal{H}$ be two DAGs over vertices $[D] \cup S$ and $[D] \cup S'$, respectively, i.e., defined over the same variables $[D]$ but possibly with different selections. Let $\mathcal{I}$ and $\mathcal{J}$ be two collections of intervention targets with a same size $1 + K$. The pairs $(\mathcal{G}, \mathcal{I})$ and $(\mathcal{H}, \mathcal{J})$ are said to be *Markov equivalent*, denoted by $(\mathcal{G}, \mathcal{I}) \sim (\mathcal{H}, \mathcal{J})$, if they imply the same set of CIs in each distribution and the same conditional invariances across distributions, as given by Theorem 1.

### 3.3.1 MAG BASICS: REPRESENTING CIS WITH LATENT AND SELECTION VARIABLES

We now introduce the MAG framework to simplify the graphical representation for the CI implications. Only the essentials are covered here; for more details, see (Richardson & Spirtes, 2002; Zhang, 2008).

A MAG is a mixed graph with three kinds of edges: directed ($\rightarrow$), bi-directed ($\leftrightarrow$), and undirected (—). Given any DAG $\mathcal{G}$ over vertices partitioned as $O$ (observed), $L$ (latent), and $S$ (selected), a corresponding MAG over $O$, denoted by $\mathcal{M}_{\mathcal{G}}$, can be constructed. Before presenting the construction rules, let us recap the motivation: to capture the CI relations among $O$ marginalized over $L$ and conditioned on $S$.

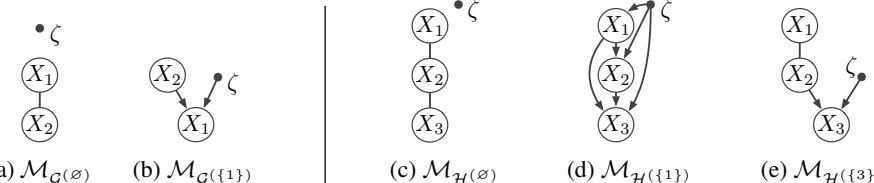

Figure 4: Examples of MAGs of interventional twin graphs. Readers may reconstruct these MAGs from DAGs in Figure 3 using either general rules (Definitions 3 and 4) or interventional twin graph-specific rules (Lemmas 4 to 6) and verify if they match. Readers may also verify if the CI implications (Theorem 1) match between d-separations on DAGs and m-separations (Definition 5) on MAGs.

First, for two observed variables $i, j \in O$, when are they always d-connected given $C \cup S$ for any subset of observed variables $C \subset O \backslash \{i, j\}$? The answer is given by the adjacencies in the MAG:

**Definition 3** (**MAG construction step 1: adjacencies; Richardson & Spirtes (2002)**). For each pair of observed variables $i, j \in O$, $i$ and $j$ are adjacent in $\mathcal{M}_\mathcal{G}$ if and only if $i, j$ are adjacent in $\mathcal{G}$, or there exists a path $p$ between $i$ and $j$ in $\mathcal{G}$ where every non-endpoint observed or selected vertex on $p$ is a collider on $p$, and every collider on $p$ is an ancestor of $i$ or $j$ or a member of $S$.

Next, we orient these adjacencies to fully capture the d-separation relations implied by $\mathcal{G}$ on $O|S$:

**Definition 4** (**MAG construction step 2: orientations; Richardson & Spirtes (2002)**). For each two adjacent vertices $i, j$ as defined by Definition 3, orient the edge in $\mathcal{M}_\mathcal{G}$ as $i \to j$ if $i \in \mathrm{an}_\mathcal{G}(\{j\} \cup S)$ and $j \notin \mathrm{an}_\mathcal{G}(\{i\} \cup S)$; $i \leftarrow j$ if $j \in \mathrm{an}_\mathcal{G}(\{i\} \cup S)$ and $i \notin \mathrm{an}_\mathcal{G}(\{j\} \cup S)$; $i \leftrightarrow j$ if $i \notin \mathrm{an}_\mathcal{G}(\{j\} \cup S)$ and $j \notin \mathrm{an}_\mathcal{G}(\{i\} \cup S)$; and $i - j$ otherwise, i.e., if $i \in \mathrm{an}_\mathcal{G}(\{j\} \cup S)$ and $j \in \mathrm{an}_\mathcal{G}(\{i\} \cup S)$.

The two steps above give general MAG construction rules. In a MAG, a vertex $j$ is a *descendant* of $i$ if $i = j$ or $i \to \cdots \to j$. An edge between $i, j$ is *into* $j$ if it is $i \to j$ or $i \leftrightarrow j$. A vertex $i$ is a *collider* on a path $p$ if both edges incident to $i$ on $p$ are into $i$. A path $p$ as $(i, k, j)$ is a *v-structure* if $i$ and $j$ are not adjacent, and $k$ is a collider on $p$. Using these notations, the MAG's CI implications are as follows:

**Definition 5** (**m-separation**). In the MAG $\mathcal{M}_\mathcal{G}$, a path $p$ between vertices $i$ and $j$ is *open* relative to a vertex set $C$ ($i, j \notin C$), if every non-collider on $p$ is not in $C$, and every collider on $p$ has a descendant in $C$. $i$ and $j$ are *m-separated* by $C$, denoted by $i \perp\!\!\!\perp_m j | C$, if there is no open path between $i, j$ relative to $C$. $i \perp\!\!\!\perp_m j | C$ holds in MAG $\mathcal{M}_\mathcal{G}$, if and only if $i \perp\!\!\!\perp_d j | C \cup S$ holds in DAG $\mathcal{G}$.

### 3.3.2 GRAPHICAL CRITERIA FOR MARKOV EQUIVALENCE

The MAG framework is useful, as it represents CIs in distributions under latent variables and selection, and determines Markov equivalence between such distributions, precisely aligning with our goal. Accordingly, we construct MAGs of interventional twin graphs, with vertices partitioned into observed ($X \cup \{\zeta\}$), latent ($X^*_{\mathrm{aff}} \cup \mathcal{E}_{\mathrm{aff}}$), and selected ($S^*$) ones. Examples of such MAGs are shown in Figure 4. While general MAG construction rules are outlined in §3.3.1, to better understand the CI implications graphically, we present the following construction rules specific to interventional twin graph.

Given a DAG $\mathcal{G}$ over $[D] \cup S$ and a target $I \subset [D]$, denote by $\mathcal{M}_{\mathcal{G}^{(I)}}$ the MAG constructed on $X \cup \{\zeta\}$ from the interventional twin graph $\mathcal{G}^{(I)}$. We show the specific rules to construct $\mathcal{M}_{\mathcal{G}^{(I)}}$, by presenting the following lemmas and the questions they aim to answer.

**Lemma 3** (**When are two variables always dependent in pure observational data?**). *For any $i, j \in [D]$, $X_i$ and $X_j$ are adjacent in $\mathcal{M}_{\mathcal{G}^{(\varnothing)}}$, if and only if $i$ and $j$ are adjacent in $\mathcal{G}$, or $\mathrm{ch}_\mathcal{G}(i) \cap \mathrm{ch}_\mathcal{G}(j) \cap \mathrm{an}_\mathcal{G}(S) \neq \varnothing$, i.e., they involve in a same selection, or have a common child that is ancestrally selected.*

The adjacencies in Lemma 3 reflect additional dependencies due to selection bias (Lemma 1). Furthermore, interventions can introduce even more dependencies, as shown in Lemma 2–again, recall the pest control example. We now generalize Lemma 3 to characterize all these dependencies.

**Lemma 4** (**When are two variables always dependent under an intervention?**). *For any $i, j \in [D]$, $X_i$ and $X_j$ are adjacent in $\mathcal{M}_{\mathcal{G}^{(I)}}$, if and only if $X_i$ and $X_j$ are adjacent in the observational MAG $\mathcal{M}_{\mathcal{G}^{(\varnothing)}}$, or there exists a path between $X_i$ and $X_j$ in $\mathcal{M}_{\mathcal{G}^{(\varnothing)}}$, where all non-endpoint vertices are affected (i.e., in $\deg_\mathcal{G}(I)$), and they, together with $i, j$, are all ancestrally selected (i.e., in $\mathrm{an}_\mathcal{G}(S)$).*

Further generalizing the adjacencies in Lemma 4 to the indicator variable $\zeta$, we have:

**Lemma 5** (**When does an intervention alters all of a variable's conditional distributions?**). *For any $j \in [D]$, $\zeta$ and $X_j$ are adjacent in $\mathcal{M}_{\mathcal{G}^{(I)}}$, if and only if $j$ is directly targeted (i.e., $j \in I$), or $j$ is both indirectly affected and ancestrally selected (i.e., $j \in \deg_\mathcal{G}(I) \cap \mathrm{an}_\mathcal{G}(S)$).*

Lemma 5 implies the unidentifiability of direct intervention targets when they are unknown. This is different from the case without selection, where "unknown target is no longer a challenge" (§2.1).

After constructing the adjacencies of $\mathcal{M}_{\mathcal{G}(I)}$ in Lemmas 3 to 5, we conclude with edge orientations:

**Lemma 6 (MAG of interventional twin graphs).** *The MAG $\mathcal{M}_{\mathcal{G}(I)}$ consists of the following edges:*

- *"Pseudo" direct intervention target edges (by Lemma 5): $\{\zeta \rightarrow X_i\}_{i \in I \cup (\deg_{\mathcal{G}}(I) \cap \mathrm{an}_{\mathcal{G}}(S))}$;*
- *Edges among $X$. Let $\mathcal{M}_{\mathcal{G}}$ be the MAG of $\mathcal{G}$ over $[D]$. For each adjacent $X_i, X_j$ (by Lemma 4):*
    - *If $i \rightarrow j$ is in the original MAG $\mathcal{M}_{\mathcal{G}}$, then orient $X_i \rightarrow X_j$ in interventional MAG $\mathcal{M}_{\mathcal{G}(I)}$;*
    - *Otherwise (i.e., $i - j \in \mathcal{M}_{\mathcal{G}}$ or $i, j$ are not adjacent in $\mathcal{M}_{\mathcal{G}}$):*
        * *If $i \notin \deg_{\mathcal{G}}(I)$ and $j \notin \deg_{\mathcal{G}}(I)$: orient $X_i - X_j$;*
        * *If $j \in \deg_{\mathcal{G}}(I)$ and $(i \notin \deg_{\mathcal{G}}(I)$ or $i \in \mathrm{an}_{\mathcal{G}}(j))$: orient $X_i \rightarrow X_j$;*
        * *If $i \in \deg_{\mathcal{G}}(I)$ and $(j \notin \deg_{\mathcal{G}}(I)$ or $j \in \mathrm{an}_{\mathcal{G}}(i))$: orient $X_j \rightarrow X_i$;*
        * *Otherwise, i.e., both $i, j \in \deg_{\mathcal{G}}(I)$, but neither is another's ancestor: orient $X_i \leftrightarrow X_j$.*

Finally, we present the graphical criteria for Markov equivalence, using MAG construction rules defined above. By enforcing same CIs in each setting and same invariances across settings, we have:

**Theorem 2 (Graphical criteria for Markov equivalence).** *For two DAGs $\mathcal{G}$ and $\mathcal{H}$ with two collections of targets $\mathcal{I} = \{I^{(0)}, I^{(1)}, \cdots, I^{(K)}\}$ and $\mathcal{J} = \{J^{(0)}, J^{(1)}, \cdots, J^{(K)}\}$, the Markov equivalence $(\mathcal{G}, \mathcal{I}) \sim (\mathcal{H}, \mathcal{J})$ holds, if and only if for each $k = 0, 1, \cdots, K$, the corresponding MAGs of interventional twin graphs, $\mathcal{M}_{\mathcal{G}^{I^{(k)}}}$ and $\mathcal{M}_{\mathcal{H}^{J^{(k)}}}$, have the same adjacencies and v-structures[4].*

Below let us examine such Markov equivalence and its graphical criteria on Examples 1 and 2:

**Example 4.** In the clinical example, let $\mathcal{G}$ be the DAG in Figure 3a and $\mathcal{G}'$ the DAG $2 \rightarrow 1$ without selection. For $\mathcal{I} = \{\varnothing, \{1\}\}$, the equivalence $(\mathcal{G}, \mathcal{I}) \sim (\mathcal{G}', \mathcal{I})$ holds, i.e., intervening on one variable cannot identify if the two variables are causally related or just correlated by selection. However, adding an intervention on the other variable can identify this distinction: $(\mathcal{G}, \mathcal{I}) \not\sim (\mathcal{G}', \mathcal{I})$, for $\mathcal{I} = \{\varnothing, \{1\}, \{2\}\}$. In the pest control example, let $\mathcal{H}$ be the DAG in Figure 3c and $\mathcal{H}'$ the DAG $1 \rightarrow 2 \rightarrow S_1 \leftarrow 3$. For $i = 1, 2$, $(\mathcal{H}, \{\varnothing, \{i\}\}) \not\sim (\mathcal{H}', \{\varnothing, \{i\}\})$, i.e., interventions on upstream variables can distinguish the two, but not on downstream $X_3$. With unknown targets, however, upstream interventions may still leave the two indistinguishable, e.g., $(\mathcal{H}, \{\varnothing, \{1\}\}) \sim (\mathcal{H}', \{\varnothing, \{1, 3\}\})$.  △

## 4 ALGORITHM: INTERVENTIONAL CAUSAL DISCOVERY UNDER SELECTION

In this section, we develop Algorithm 1, named Causal Discovery from Interventional data under potential Selection bias (CDIS). Using the twin graph framework and Markov properties from §3, this algorithm learns causal relations and selection structures up to the equivalence class, from interventional data with soft interventions, unknown targets, and potential selection bias. We assume causal sufficiency and faithfulness, i.e., no CIs beyond those implied by the graph (Theorem 1).

A first thought might be to obtain adjacencies from observational data $p^{(0)}$, as it provides the sparsest skeleton (Lemmas 1 and 2). Then, one could form v-structures involving intervention indicators by checking conditional (in)variances, and use these v-structures for further orientation on the sparsest skeleton. However, as we will show, this seemingly intuitive approach can lead to false discoveries:

**Example 5.** Consider the DAG $\mathcal{G} = 1 \rightarrow 2 \rightarrow S_1 \leftarrow 3$ with targets $\mathcal{I} = \{\varnothing, \{1\}\}$. CIs in $p^{(0)}$ first yield a skeleton $1 - 2 - 3$, and the target is identified as $\zeta \rightarrow \{1, 2\}$. Since $p(X_3)$ does not change ($\zeta \perp\!\!\!\perp X_3$), the v-structure $\zeta \rightarrow 2 \leftarrow 3$ is formed. Now, if we directly apply orientation rules (Meek, 1995) on the skeleton, the non-collider $1 \leftarrow 2 \leftarrow 3$ will be oriented, leading to a false edge $1 \leftarrow 2$.  △

Example 5 highlights the pitfalls of directly applying orientations on adjacencies obatined from pure observational data, even if they are the sparsest and closest to truth. Instead, orientations must be applied on denser adjacencies from interventional data to prevent false propagation, and then used to refine edges in the sparsest skeleton. Our CDIS algorithm is built on this principle.

The pseudocode for CDIS is detailed in Algorithm 1 in Appendix B due to page limit. Below we provide a high-level summary. CDIS consists of the following three steps:

**Step 1. Maximal orientation from pure observational data.** Run FCI (Zhang, 2008) on $p^{(0)}$, we obtain the maximal information possible from $p^{(0)}$ only, represented by a PAG [5] $\hat{\mathcal{M}}^{(0)}$.

---

[4]Another condition for general MAG equivalence is not needed here, due to twin graphs' specific structures.

[5]Partial ancestral graph (PAG) is a graph to represent a class of MAGs. See definitions in Appendix B.

**Step 2. Maximal orientation from interventional data.** For each $k = 1, \cdots, K$, orientations are derived from pooled data $(p^{(0)}, p^{(k)})$, represented by a PAG $\hat{\mathcal{M}}^{(k)}$ over $[D] \cup \{\zeta\}$. Significant pruning is applied since conditional dependencies from $p^{(0)}$ must hold in $p^{(k)}$.

**Step 3. Refinement using interventional twin graph-specific criteria.** Guided by the specific construction rules in Lemmas 4 to 6, information from $\hat{\mathcal{M}}^{(k)}$ are used to orient uncertain edges in $\hat{\mathcal{M}}^{(0)}$, i.e., to narrow down the equivalence class about the true model. Updated $\hat{\mathcal{M}}^{(0)}$ then guides further refinement on $\hat{\mathcal{M}}^{(k)}$, iterating until no new orientations are possible.

We show that CDIS correctly identifies and distinguishes causal relations and selection mechanisms:

**Theorem 3** (**Soundness of CDIS**). *Let $\hat{\mathcal{M}}^{(0)}$ be the output PAG of Algorithm 1 with oracle CI tests on interventional data $\{p^{(k)}\}_{k=0}^K$ given by $(\mathcal{G}, \mathcal{I})$. Then, $\hat{\mathcal{M}}^{(0)}$ is consistent with MAG $\mathcal{M}_{\mathcal{G}}$. Specifically,*

1. *For any $i \to j \in \hat{\mathcal{M}}^{(0)}$, there is also $i \to j \in \mathcal{G}$, and $j$ is not ancestrally selected in $\mathcal{G}$.*
2. *For any $i - j \in \hat{\mathcal{M}}^{(0)}$, both $i$ and $j$ are ancestrally selected in the true DAG $\mathcal{G}$.*

The soundness of CDIS follows from the soundness of FCI rules, but its completeness–whether all invariant edges in the equivalence class are identified–remains uncertain to us. Through brutal search on $\sim 50,000$ $(\mathcal{G}, \mathcal{I})$ pairs with up to 15 vertices, we have not found any incomplete example, and thus we conjecture its completeness. However, proving this is challenging, since the refining step relies on our graphical criteria (Lemma 6) as background knowledge, for which no complete rules exist yet (Andrews et al., 2020; Wang et al., 2024; Venkateswaran & Perković, 2024). This technical difficulty mirrors also to FCI itself: though FCI has been widely adopted since Spirtes et al. (1999), its completeness result (without background knowledge) was not established until Ali et al. (2005) (partially) and Zhang (2008) (fully). Notably, in our scenario, FCI is guaranteed complete on pure observational data, while CDIS provides more information than that.

## 5 EXPERIMENTS AND RESULTS

In this section, we present empirical studies on simulations and real-world data to demonstrate that our algorithm effectively identifies true causal relations despite the presence of selection bias.

### 5.1 SIMULATIONS

We conduct simulations to validate the soundness of our proposed method, CDIS, and to explore the validity of its conjectured completeness. We compare CDIS against existing methods such as GIES (Hauser & Bühlmann, 2012), IGSP (Wang et al., 2017), UT-IGSP (Squires et al., 2020), CD-NOD (Huang et al., 2020), and the JCI-GSP used in Squires et al. (2020) as an extension of JCI (Mooij et al., 2020) with GSP (Solus et al., 2021). Further details are described in Appendix E.

We follow the data generating procedure outlined in Definition 1. Specifically, we begin by randomly sampling Erdös–Rényi (Erdös & Rényi, 1959) graphs with an average degree of 2 as the ground truth DAG for $\{X_i^*\}_{i=1}^D$. Next, to represent how multiple selection mechanisms operate simultaneously, we introduce $\lfloor D/5 \rfloor$ selection variables $S_1^*, \ldots, S_{\lfloor D/5 \rfloor}^*$, where each $S_i^*$ is randomly assigned one or two parent variables from $\{X_i^*\}_{i=1}^D$. Note that we do not allow the last $\lfloor D/2 \rfloor$ variables in the causal ordering to be involved in selection, because doing so could result in the corresponding PAGs having an excessive number of '∘—∘' edges, making them overly uninformative.

We then simulate linear SEMs for $\{X_i^*\}_{i=1}^D$ with exogenous noise terms $\{\epsilon_i^*\}_{i=1}^D$. Sample selection is performed based on each $S_i^*$, as a linear sum of its parents in $X^*$ and an independent noise, falling within a predefined interval and ensuring the desired sample size (5,000 samples after selection). Here, the linear edge coefficients are sampled from $\mathrm{Unif}([-2, -0.5] \cup [0.5, 2])$, and the variances of exogenous noise terms are sampled from $\mathrm{Unif}[1, 4]$. Finally, using the exogenous noise terms $\{\epsilon_i^*\}_{i=1}^D$ of the selected samples, we simulate interventional data over $\{X_i\}_{i=1}^D$ under different targets. We simulate a total of $\lfloor D/2 \rfloor$ interventions, each with one variable being intervened on.

In order to show how CDIS reliably identifies correct causal relations, we report the precision, recall, and F1 score of the PAG '→' edges estimated on the simulated data, as compared to the true '→' edges that can be theoretically discovered, indicated by Theorem 3. Furthermore, we report the accuracy of the edgemark of the estimated PAGs, as well as the structural Hamming distance (SHD) of its corresponding skeleton, as compared to the true PAGs. For the CPDAGs estimated by the baselines, we treat their directed and undirected edges as '→' and '∘—∘' edges, respectively.

The three main metrics are given in Figure 5, while the remaining ones are available in Figure 12 in Appendix E. The experimental results demonstrate that CDIS[6] outperforms the baselines across most settings and metrics, especially for precision. Notably, the average precision of our algorithm exceeds 0.7 across most configurations. In contrast, the baselines generally achieve precision below 0.65 (with the exception of UT-IGSP in some instances). These observations validate the soundness of CDIS in identifying true causal relations, while suggesting that other methods may falsely infer spurious causal relations, possibly due to selection bias, as illustrated in Examples 1 and 2.

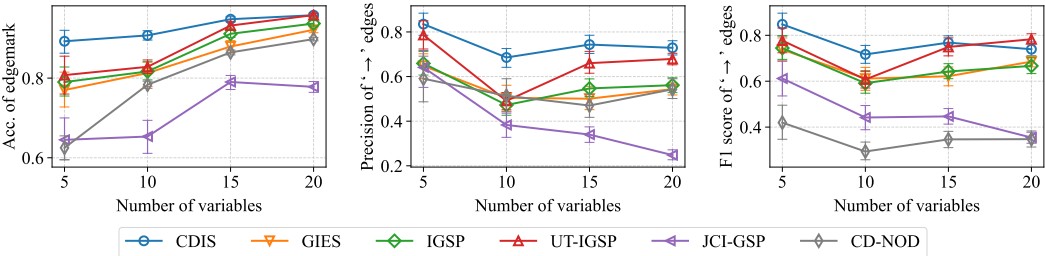

Figure 5: Empirical results across different numbers of variables. Here, the first figure shows the accuracy of the edgemark of the estimated PAGs. The rest two figures show the precision and F1 score of the '→' edges in PAGs. The error bars illustrate the standard errors from 20 random simulations.

## 5.2 REAL-WORLD APPLICATIONS

We evaluate the gene regulation networks (GRNs) of 24 previous reported essential regulatory genes encoding different transcription factors (TFs) (Yang et al., 2018) using a single-cell perturbation data, i.e., sciPlex2 (Peidli et al., 2024), where the A549 cells, a human lung adenocarcinoma cell line, are either exposed to one of the four different transcription modulators, including dexamethasone, nutlin-3a, BMS-345541, and suberoylanilide hydroxamic acid (SAHA), or simply treated with dimethylsulfoxide vehicle as control (Srivatsan et al., 2020). As shown in Figure 9, we discovered some validated regulatory relationships like *RELA → RUNX1* and *JUNB → MAFF*, since RELA has been implicated as a *RUNX3* transcription regulator (Zhou et al., 2015) and *Maf* family was upregulated by *JunB* (Koizumi et al., 2018), despite the extensive co-regulation between these two genes (Kataoka et al., 1994). A more detailed analysis of these results is provided in Appendix E.

We also apply CDIS to an educational dataset (Table 1), from a random controlled trial evaluating the effects of incentives and services on college freshmen's academic achievements (Angrist et al., 2009). First-year undergraduates are randomly assigned to either the control group or one of the three treatment arms: a service strategy (Student Support Program, SSP) with both peer-advising service and supplemental instruction service in facilitated study groups; an incentive strategy (the Student Fellowship Program, SFP) with the opportunity to win merit scholarships based on academic achievements; and an intervention offering both named SFSP. As depicted in Figure 10, subgroup analysis stratified by genders indicate that SSP only improves the women's performance while SFP shows effects only on men (see Figure 11). The results indicates that interventions exhibit genuine heterogeneous treatment effects on college students' academic performance, with the moderating effect of gender rather than being attributed to selection bias based on gender.

## 6   CONCLUSION AND LIMITATIONS

We introduce a new problem setup for interventional causal discovery with selection bias. We show how and why existing models fail to represent the data, propose a new model to capture intervention and selection dynamics, characterize Markov properties and a criterion for equivalence, and develop a sound algorithm called CDIS for identifying causal relations and selection mechanisms.

One could naturally extend the graph by introducing $S$ also to the reality world to model a new round of post-intervention selection, e.g., *lost to follow-up* (Akl et al., 2012). Another extension can be the causal inference results based on our model. While we provide the graphical criteria to determining equivalence, a graphical representation for the equivalence class is to be developed. Also, the completeness guarantee of the CDIS algorithm, though hypothesized, is yet to be proven.

---

[6]A Python implementation of CDIS is available at `https://github.com/MarkDana/CDIS`.

ACKNOWLEDGMENTS

We would like to acknowledge the support from NSF Award No. 2229881, AI Institute for Societal Decision Making (AI-SDM), the National Institutes of Health (NIH) under Contract R01HL159805, and grants from Quris AI, Florin Court Capital, and MBZUAI-WIS Joint Program. ZT is supported by the National Institute of Justice (NIJ) Graduate Research Fellowship, Award No. 15PNIJ-24-GG-01565-RESS.

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

## A    PROOFS OF MAIN RESULTS

Since the technical development of this paper is fundamentally built upon the characterization of maximal ancestral graphs (MAGs), we focus on proving only those results that do not directly follow from the basic properties of MAGs. Specifically, the results stated in Theorem 1, Lemma 1, Lemma 2, Lemma 3, and Lemma 6 are immediate consequences of the definition of MAGs(Richardson & Spirtes, 2002; Zhang, 2008). Therefore, their proofs are omitted.

**Lemma 4 (When are two variables always dependent under an intervention?).** *For any $i, j \in [D]$, $X_i$ and $X_j$ are adjacent in $\mathcal{M}_{\mathcal{G}(I)}$, if and only if $X_i$ and $X_j$ are adjacent in the observational MAG $\mathcal{M}_{\mathcal{G}(\varnothing)}$, or there exists a path between $X_i$ and $X_j$ in $\mathcal{M}_{\mathcal{G}(\varnothing)}$, where all non-endpoint vertices are affected (i.e., in $\deg_{\mathcal{G}}(I)$), and they, together with $i, j$, are all ancestrally selected (i.e., in $\mathrm{an}_{\mathcal{G}}(S)$).*

*Proof of Lemma 4.* For any $i, j \in [D]$, $X_i$ and $X_j$ are adjacent in the MAG $\mathcal{M}_{\mathcal{G}(I)}$, if and only if there is an inducing path between $X_i$ and $X_j$ in the interventional twin DAG $\mathcal{G}^{(I)}$. We now consider this DAG $\mathcal{G}^{(I)}$. We consider by cases where $i, j$ are affected by the intervention or not.

1. Consider cases where both $i$ and $j$ are not affected, i.e., $X_i$ and $X_j$ are not splited in the DAG $\mathcal{G}^{(I)}$. When $X_i$ and $X_j$ are adjacent in the observational MAG $\mathcal{M}_{\mathcal{G}(\varnothing)}$, according to Lemma 3, either $i$ and $j$ are adjacent in $\mathcal{G}$ or they have a common child that is in $\mathrm{an}_{\mathcal{G}}(S)$. Therefore, they still have this inducing path in between in the DAG $\mathcal{G}^{(I)}$. When the two are not adjacent themselves, but there is a path $(X_i, X_{l_1}, \cdots, X_{l_m}, X_j)$ in $\mathcal{M}_{\mathcal{G}(\varnothing)}$ where $\{i, l_1, \cdots, l_m, j\} \subset \mathrm{an}_{\mathcal{G}}(S)$ and $\{l_1, \cdots, l_m\} \subset \deg_{\mathcal{G}}(I)$: then in the DAG $\mathcal{G}^{(I)}$ all of $\{l_1, \cdots, l_m\}$ are splited into $\{X^*_{l_1}, \cdots, X^*_{l_m}\}$. For each two adjacent vertices on this path, we know that are either adjacent or have a common child that is an ancestor of $S^*$ in $\mathcal{G}^{(I)}$. We can then consider this path connecting them on $\mathcal{G}^{(I)}$ (note that we can safely assume the common children, if any, are different to each other, as if, for example, $l_1, l_2$ and $l_2, l_3$ have a same common child to render them adjacent, then we can always build the path $(l_1, l_3, \cdots)$ to bypass $l_2$, where $l_1, l_3$ must also be adjacent in $\mathcal{M}_{\mathcal{G}(\varnothing)}$). On this path, except for the two end $X_i$ and $X_j$, all the vertices are not observed. All the colliders must be ancestor of $S^*$. Therefore, this is an inducing path between $X_i$ and $X_j$ in $\mathcal{G}^{(I)}$.

   In above, we have proved the '$\Leftarrow$' direction. For the '$\Rightarrow$'direction, consider the inducing path between $X_i$ and $X_j$. When there are more than two nontrivial vertices on it, they must be all in $X^*$, i.e., all the intermediate vertices are affected and splited. Otherwise, suppose there is an observed $X_c$ on the path. For it to be a valid inducing path, $X_c$ has to be a collider, $X_i \cdots \rightarrow X_c \leftarrow \cdots X_j$. If this path consists of only $X_i \rightarrow X_c \leftarrow X_j$, $X_c$ must be ancestrally selected, leading to the case of adjacent $X_i, X_j$ in $\mathcal{M}_{\mathcal{G}(\varnothing)}$. Otherwise, when there are more than $X_c$ on the path, the intermediate vertex next to $X_c$ must also not be affected (and is thus observed), as it is a parent of $X_c$. However, this observed variable cannot be a collider, making this not an inducing path anymore. Similarly, we can show all the intermediate vertices and $X_i, X_j$ must be ancestrally selected.

2. Consider cases where both $i$ and $j$ are affected, i.e., $X_i$ and $X_j$ are both splited in the DAG $\mathcal{G}^{(I)}$. Consider the path $(X_i \leftarrow \epsilon_i \rightarrow X^*_i, X^*_{l_1}, \cdots, X^*_{l_m}, X^*_j \leftarrow \epsilon_j \rightarrow X_j)$ in the DAG $\mathcal{G}^{(I)}$, all the colliders must be in the '$*$' part, then similar to 1., this is also be an inducing path.

   For the '$\Rightarrow$' direction, since the right reality world is not ancestor of the $S^*$ variables, the inducing path, if any, must route back to the '$*$' world (otherwise, there are observed but non-collider vertices). The rest is similar to that of [1.].

3. For cases where one of $i$ and $j$ is affected and the other is not, similar to 1. and 2., the inducing path can also be built. Also, the '$\Rightarrow$' direction goes in a same way.

$\square$

**Lemma 5** (**When does an intervention alters all of a variable's conditional distributions?**). *For any $j \in [D]$, $\zeta$ and $X_j$ are adjacent in $\mathcal{M}_{\mathcal{G}^{(I)}}$, if and only if $j$ is directly targeted (i.e., $j \in I$), or $j$ is both indirectly affected and ancestrally selected (i.e., $j \in \deg_{\mathcal{G}}(I) \cap \mathrm{an}_{\mathcal{G}}(S)$).*

*Proof of Lemma 5.*

1. (**'$\Leftarrow$' direction**)
   - If $j \in I$, there is a direct edge $\zeta \to X_j$ in $\mathcal{G}^{(I)}$, and so this directed edge is also in $\mathcal{M}_{\mathcal{G}^{(I)}}$. Hence, $\zeta$ cannot be m-separated from $X_j$ given any $X_C$.
   - Otherwise, $j \in \deg_{\mathcal{G}}(I) \cap \mathrm{an}_{\mathcal{G}}(S)$. Consider the directed path from $\zeta$ to $X_j$ on the DAG $\mathcal{G}^{(I)}$. For any $X_C \subset X \backslash \{X_j\}$,
     * If $X_C$ contains no vertices on this path, then this path remains open, and thus $\zeta$ and $X_j$ are not d-separated by $X_C$ and $S^*$.
     * Otherwise, let $X_l$ be the first vertex on this path that is in $X_C$. Consider the path $\zeta \to \cdots \to X_l \leftarrow \epsilon_l \to X_l^* \to \cdots X_j^* \leftarrow \epsilon_j \to X_j$. The collider $X_l$ is conditioned on, and the collider $X_i^*$ has its descendant in $S^*$ conditioned on. All the noncolliders from $\zeta$ to $X_l$ are not conditioned on. All the noncolliders from $X_l^*$ to $X_j^*$ are either not conditioned on (since they are changed by intervention). Therefore, this path remains open, and thus $\zeta$ and $X_j$ are not d-separated by $X_C$ and $S^*$.
2. (**'$\Rightarrow$' direction**) By contradiction, suppose $j \notin I \cup (\deg_{\mathcal{G}}(I) \cap \mathrm{an}_{\mathcal{G}}(S))$.
   - If $j \notin \deg_{\mathcal{G}}(I)$, i.e., intervention $\zeta$ doesn't change $X_j$, then $X_j$ and $\zeta$ are d-separated by empty set ($X_j$ is not even splited). Contradiction.
   - Otherwise, $j \in \deg_{\mathcal{G}}(I)$, but $j$ is not an ancestor of any $S$ in $\mathcal{G}$. Let $C$ be $\mathrm{pa}_{\mathcal{G}}(i)$. We now show that this $X_C$ m-separates $\zeta$ from $X_j$. Consider any path $p$ between $\zeta$ and $X_j$ in $\mathcal{G}^{(I)}$.
     * If $p$ contains no vertices in the counterfactual $*$ world, then by Markov condition ($X_j$ is d-separated from its non-descendant $\zeta$ by its parents), $p$ is blocked by $X_C$.
     * Otherwise, let $X_l$ be the last vertex on $p$ that comes out from the counterfactual world, i.e., $p$ is in the form of $\zeta \to \cdots - X_l^* \leftarrow \epsilon_l \to X_l - \cdots - X_j$. For this path to be open, at least the part between $X_l$ and $X_j$ should be open. So, similarly by the Markov condition, $X_l$ must be $X_j$'s descendant. Now obviously on $p$ between $\zeta \to$ and $\leftarrow \epsilon_l$ there must be a collider. Let $A^*$ be first collider counting from $\epsilon_l$, i.e., $\cdots \to A^* \leftarrow \cdots \leftarrow X_l^* \leftarrow \epsilon_l$. We use $A^*$ because it must be in the counterfactual world, and can be either among $X^*$ or $S^*$. $A^*$ can be $X_l^*$ itself. To remain $p$ open, $A^*$ has to be an ancestor of $S^*$. So $X_l^*$ is an ancestor of $S^*$. So $X_j^*$ is an ancestor of $S^*$, contradicted to our presumption that $j \notin \mathrm{an}_{\mathcal{G}}(S)$. So such an open $p$ does not exist.

$\square$

**Theorem 2** (**Graphical criteria for Markov equivalence**). *For two DAGs $\mathcal{G}$ and $\mathcal{H}$ with two collections of targets $\mathcal{I} = \{I^{(0)}, I^{(1)}, \cdots, I^{(K)}\}$ and $\mathcal{J} = \{J^{(0)}, J^{(1)}, \cdots, J^{(K)}\}$, the Markov equivalence $(\mathcal{G}, \mathcal{I}) \sim (\mathcal{H}, \mathcal{J})$ holds, if and only if for each $k = 0, 1, \cdots, K$, the corresponding MAGs of interventional twin graphs, $\mathcal{M}_{\mathcal{G}^{I^{(k)}}}$ and $\mathcal{M}_{\mathcal{H}^{J^{(k)}}}$, have the same adjacencies and v-structures[7].*

*Proof of Theorem 2.* The Markov equivalence $(\mathcal{G}, \mathcal{I}) \sim (\mathcal{H}, \mathcal{J})$ holds if and only if the MAGs on $X \cup \{\zeta\}$ of each interventional twin graphs $\mathcal{G}^{I^{(k)}}$ and $\mathcal{H}^{J^{(k)}}$ encode the same m-separations. According to the MAG characterization, this means that they have the same adjacencies, the same v-structures, and if a path $p$ is a discriminating path for a vertex $V$ in both graphs, then $V$ is a collider on the path in one graph if and only if it is a collider on the path in the other. Our condition covers the first two cases, and now we show that we do not need to consider the third case.

We show that in any discriminating path $(X, W_1, \cdots, W_m, V, Y)$ in a MAG of a interventional twin graph (for detailed definition about the discriminating path, please refer to Definition 7 in (Zhang, 2008). Here we abuse the notation $X$ to denote one vertex, to be consistent with the definitions there), $V$ must not be a collider, i.e., it must be $V \to Y$. Suppose that $V$ is a collider, it must be $\cdots \to W_m \leftrightarrow V \leftrightarrow Y$, as otherwise there will be an almost directed cycle $W \to Y \to V \leftrightarrow W_m$. Then, according to Lemma 6, with those '$\leftrightarrow$' edges, all of $W_1, \cdots, W_m, V, Y$ are ancestrally selected, and are influenced by the intervention. Consider two cases:

---

[7] Another condition for general MAG equivalence is not needed here, due to twin graphs' specific structures.

1° If $X$ is ancestrally selected, according to Lemma 4, the path $(X, W_1, \cdots, W_m, V, Y)$ will render $X$ and $Y$ induced adjacent, violating the definition for the discriminating path.

2° Otherwise ($X$ is not ancestrally selected), it must be a true directed edge $X \to Y$ in the original $\mathcal{G}$, as the condition for (nontrivial) induced adjacency is violated. Then, since $W_1$ is already ancestrally selected, it is impossible for $X$ to be not selected.

Then, we have shown that all $V$ nodes on discriminating paths must be non-colliders. Therefore, the equivalence of MAGs is given by the first two conditions: same adjacencies and v-structures. $\qquad\square$

**Theorem 3 (Soundness of CDIS).** *Let $\hat{\mathcal{M}}^{(0)}$ be the output PAG of Algorithm 1 with oracle CI tests on interventional data $\{p^{(k)}\}_{k=0}^{K}$ given by $(\mathcal{G}, \mathcal{I})$. Then, $\hat{\mathcal{M}}^{(0)}$ is consistent with MAG $\mathcal{M}_{\mathcal{G}}$. Specifically,*

1. *For any $i \to j \in \hat{\mathcal{M}}^{(0)}$, there is also $i \to j \in \mathcal{G}$, and $j$ is not ancestrally selected in $\mathcal{G}$.*
2. *For any $i \,\text{—}\, j \in \hat{\mathcal{M}}^{(0)}$, both $i$ and $j$ are ancestrally selected in the true DAG $\mathcal{G}$.*

*Proof of Theorem 3.* The orientation rules of Algorithm 1 directly follow from the characterization of MAGs in Lemma 6. The soundness of the algorithm follows from the soundness of FCI's orientation rules. The only aspect requiring further elaboration is the use of "$\texttt{FCI}^+$," where all $\circ\!\!\to$ edges are further oriented as $\to$ edges. This step relies on the fact that the PAGs contains no $\leftrightarrow$ edges.

For the purely observational PAG, this follows straightforwardly from the fact that in the MAG $\mathcal{M}_{\mathcal{G}(\varnothing)}$, there are no bidirected edges, as our framework assumes no latent variables beyond those indexed by $[D]$. However, the case of PAGs derived from the MAGs $\mathcal{M}_{\mathcal{G}(I)}$ needs additional explanation.

According to Lemma 6, the corresponding MAGs $\mathcal{M}_{\mathcal{G}(I)}$ may indeed contain $\leftrightarrow$ edges. However, we show that such edges cannot be identified in the PAG.

Suppose there exists an edge $X_i \leftrightarrow X_j$ in the MAG $\mathcal{M}_{\mathcal{G}(I)}$, according to Lemma 6's characterization on bidirected edges, there must be $X_i \leftarrow \zeta \to X_j$. For the arrowheads to be identifiable in the PAG, another v-structure must be present.

Assume, for instance, that there is also an edge $X_k \leftrightarrow X_j$. Then, by Lemma 4, the existence of an inducing path through $X_j$ implies that $X_i$ must also be adjacent to $X_k$ in the MAG. This results in a triangle $(X_i, X_j, X_k)$ whose orientation remains unidentifiable.

Consider the other scenario where $X_k \to X_j$. In this case, $X_k$ is also reachable from $X_j^*$ and subsequently from $X_i^*$ to $X_i$. This inducing path still renders $X_k$ and $X_i$ adjacent, so as to form a triangle where definitive determination of orientation becomes impossible.

$\qquad\square$

## B    PSEUDOCODE FOR THE CDIS ALGORITHM

Before detailing CDIS, we introduce key notations. Like a CPDAG for DAGs, we use a partial ancestral graph (PAG (Spirtes et al., 2000)) to represent a class of MAGs with same adjacencies. A PAG is a graph with six kinds of edges ( $\text{—}$ , $\to$, $\leftrightarrow$, $\circ\!\!-$, $\circ\!\!-\!\!\circ$, $\circ\!\!\to$ ) where non-circle marks indicate marks shared by all MAGs in the class. FCI (Zhang, 2008) is an algorithm that identifies the true PAG from data. In the pseudocode, FAS (fast adjacency search) denotes a function that takes in data and returns a PAG. Adjacencies are determined by CIs, with orientations as $i\circ\!\!\to k\leftarrow\!\!\circ j$ for v-structures and $i\circ\!\!-\!\!\circ j$ otherwise[8]. $\texttt{FCI}^+$ denotes a function that takes in a PAG, recursively applies the ten FCI rules and an extra rule to orient all $\circ\!\!\to$ as $\to$, and returns the PAG when no further orientations can be made.

The pseudocode for the CDIS algorithm is provided below:

---

[8]FAS can be implemented by the PC algorithm followed by an additional step of skeleton refinement. For further details, see (Spirtes et al., 2000), p. 144.

---

**Algorithm 1:** C̲ausal D̲iscovery from I̲nterventional data under potential S̲election bias (CDIS)

---

**Input:** Observational and interventional data $\{p^{(k)}\}_{k=0}^K$ over $X_{[D]}$ with unknown targets.

**Output:** A partially ancestral graph (PAG) over vertices $[D]$.

**Step 1: Get maximal orientation from pure observational data.** $\hat{\mathcal{M}}^{(0)} \leftarrow \texttt{FCI}^+(\texttt{FAS}(p^{(0)}))$.

**Step 2: Get MAG adjacencies from interventional data. for** $k \leftarrow 1, \cdots, K$ **do**

$\quad$ $\hat{\mathcal{M}}^{(k)} \leftarrow \texttt{FAS}(p^{(0)}, p^{(k)})$, running PC on pooled data of $p^{(0)}$ and $p^{(k)}$ over $\{\zeta\} \cup [D]$, with CIs
$\quad$ in forms of Theorem 1. Pruning can be made: any adjacency in $\hat{\mathcal{M}}^{(0)}$ must appear in $\hat{\mathcal{M}}^{(k)}$.

**Step 3: Refine $\hat{\mathcal{M}}^{(0)}$ using graphical criteria on MAGs of twin graphs. repeat**

$\quad$ **Step 3.1: Orient $\hat{\mathcal{M}}^{(k)}$ based on current knowledge. for** $k \leftarrow 1, \cdots, K$ **do**

$\quad\quad$ **foreach** $i \rightarrow j \in \hat{\mathcal{M}}^{(0)}$ **do** Orient $i \rightarrow j$ in $\hat{\mathcal{M}}^{(k)}$, as suggested by Lemma 6;
$\quad\quad$ **foreach** $i$ adjacent to $\zeta$, **do** Orient $\zeta \rightarrow i$ in $\hat{\mathcal{M}}^{(k)}$, as intervention indicator is exogenous;
$\quad\quad$ $\hat{\mathcal{M}}^{(k)} \leftarrow \texttt{FCI}^+(\hat{\mathcal{M}}^{(k)})$, further orientation with above background knowledge edges;

$\quad$ **Step 3.2: Update $\hat{\mathcal{M}}^{(0)}$ using information from $\hat{\mathcal{M}}^{(k)}$. foreach** *adjacent* $i, j$ in $\hat{\mathcal{M}}^{(0)}$ **do**

$\quad\quad$ **if** $\exists k$ such that $i - j \in \hat{\mathcal{M}}^{(k)}$ **then** Orient $i - j$ in $\hat{\mathcal{M}}^{(0)}$;
$\quad\quad$ **if** $\exists k$ such that $i \rightarrow j \in \hat{\mathcal{M}}^{(k)}$ and $i \circ\!\!-\! j \in \hat{\mathcal{M}}^{(0)}$ **then** Orient $i - j$ in $\hat{\mathcal{M}}^{(0)}$;
$\quad\quad$ **if** $\exists k$ such that $i \rightarrow j \in \hat{\mathcal{M}}^{(k)}$ and $p^{(0)}(X_i) \neq p^{(k)}(X_i)$ **then** Orient $i \rightarrow j$ in $\hat{\mathcal{M}}^{(0)}$;
$\quad\quad$ **if** $\exists k_1 \neq k_2$ such that $i \rightarrow j \in \hat{\mathcal{M}}^{(k_1)}$ and $i \leftarrow j \in \hat{\mathcal{M}}^{(k_2)}$ **then** Orient $i - j$ in $\hat{\mathcal{M}}^{(0)}$;

$\quad$ **Step 3.3: Further orient $\hat{\mathcal{M}}^{(0)}$ based on current knowledge.** $\hat{\mathcal{M}}^{(0)} \leftarrow \texttt{FCI}^+(\hat{\mathcal{M}}^{(0)})$;

**until** *no further orientations are found for $\hat{\mathcal{M}}^{(0)}$ in Step 3.2*;

**return** $\hat{\mathcal{M}}^{(0)}$

---

## C   MORE ELABORATIONS ON EXAMPLES AND MOTIVATIONS

### C.1   SIMULATION FOR THE PEST CONTROL EXAMPLE 2

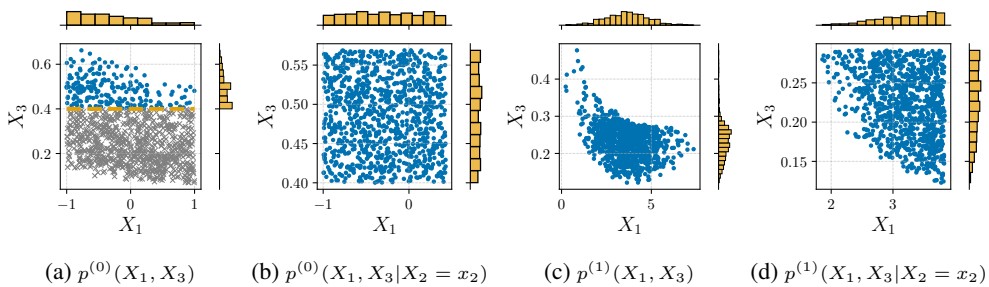

(a) $p^{(0)}(X_1, X_3)$ $\qquad$ (b) $p^{(0)}(X_1, X_3 | X_2 = x_2)$ $\qquad$ (c) $p^{(1)}(X_1, X_3)$ $\qquad$ (d) $p^{(1)}(X_1, X_3 | X_2 = x_2)$

Figure 6: (a) shows the scatterplot of $X_1; X_3$ in general population (both '×' and '•'), with only '•' individuals involved into study as $p^{(0)}$. (b) shows the scatterplot of $X_1, X_3$ in $p^{(0)}$ with $X_2$ conditioned on its mean value $x_2$, illustrating the conditional independence $X_1 \perp\!\!\!\perp X_3 | X_2$ in $p^{(0)}$. Applying an intervention to $X_1$ on the '•' individuals in (a), we get the interventional distribution $p^{(1)}$. (c) shows the scatterplot of $X_1; X_3$ in $p^{(1)}$. (d) shows the scatterplot of $X_1, X_3$ in $p^{(0)}$ with $X_2$ conditioned on its mean value $x_2$, illustrating that the intervention destroys the original condition independence, i.e., $X_1 \not\perp\!\!\!\perp X_3 | X_2$ holds in $p^{(1)}$.

Following Example 2, let the DAG $\mathcal{G}$ be $1 \rightarrow 2 \rightarrow 3 \rightarrow S_1$. Data is simulated as follows:

$$E_1, E_2, E_3 \sim \mathcal{U}[-1, 1], \text{ independently;}$$
$$X_1 = E_1;$$
$$X_2 = \texttt{sigmoid}(X_1 + E_2);$$
$$X_3 = \texttt{sigmoid}(-2X_2 + E_3);$$
$$S_1 = \mathbb{1}(X_3 > 0.4).$$

Only individuals with $S_1 = 1$ values are involved into the study and get observed, corresponding to the '•' markers in $p^{(0)}$ in Figure 6a. For these individuals, an intervention to lift $X_1$ is made:

$$X_1 = X_1 + E', \text{with } E' \sim \mathcal{N}(4, 1).$$

That is, individuals are expected to gets a lift of $4$ in its $X_1$ values, while a variance of $1$ is also given to model the randomness in applying real interventions. $X_2, X_3$ are then generated using the same generating functions and their same $E_2, E_3$ values as above. The scatterplots of the resulting interventional distribution $p^{(1)}$ are shown in Figure 6c.

To illustrate the condition independence relation $X_1 \perp\!\!\!\perp X_3 | X_2$, we show in Figure 6b and Figure 6d the scatterplots of $X_1, X_3$ with $X_2$ conditioned on their specific mean values in $p^{(0)}$ and $p^{(1)}$, respectively. Clearly, $X_1 \perp\!\!\!\perp X_3 | X_2$ holds in $p^{(0)}$ (though to be rigorous, the scatterplot on a single conditioned $x_2$ value is not enough), while this condition independence no longer holds in the interventional data $p^{(1)}$.

## C.2 WHY NOT SPLIT EVERY VARIABLE INTO COUNTERFACTUAL AND REALITY VERTICES?

In our definition of the interventional twin graph (Definition 1), the graph is defined for each single intervention with target $I$, instead of on a collections of targets $\mathcal{I}$. This is different from the usual augmented graph setting where only one augmented graph is defined, with multiple exogenous intervention indicators $\zeta_1, \cdots, \zeta_K$. The specific reason why we do not choose to use only one combined graph is that, our definition of the graph depends on each specific $I$, namely, only variables that are affected from an intervention target $I$ are split. Questions then naturally arise: can we split all variables into the two worlds, as in typical twin graph models (Balke & Pearl, 1994)? In this way, can we formulate a single graph that represents for the whole $\mathcal{I}$, which seems simpler?

In what follows, we show that a single graph with all vertices split is doable. However, in contrast to our expectation, it introduces more unnecessary complexities.

First we define this alternative model. Let $\mathbf{1}$ and $\mathbf{0}$ be vectors of all 1s and all 0s, respectively, and $\mathbb{1}_k$ be the vector with 1 at its $k$-th entry and 0s elsewhere (by convention, $\mathbb{1}_0 = \mathbf{0}$).

**Definition 6 (Alternative one-for-all interventional twin graph).** For a DAG $\mathcal{G}$ over $[D] \cup S$ and a collection of targets $\mathcal{I} = \{I^{(k)}\}_{k=0}^K$, the *alternative one-for-all interventional twin graph* $\mathcal{G}^{\mathcal{I}}$ is a DAG with vertices $X^* \cup S^* \cup \mathcal{E} \cup X \cup \zeta$, where:

- $\zeta = \{\zeta_k\}_{k \in [K]}$ are intervention indicators: $\zeta = \mathbb{1}_k$ denotes a sample from the $k$-th interventional distribution $p^{(k)}$. Specifically, $\zeta = \mathbb{1}_0 = \mathbf{0}$ denotes for the pure observational samples from $p^{(0)}$.
- $X = \{X_i\}_{i=1}^D$ are variables in the observed *reality world*, pure observational or interventional;
- $X^* = \{X_i^*\}_{i=1}^D$ and $S^* = \{S_i^*\}_{i=1}^T$ are variables in the unobserved *counterfactual basal world*, representing the corresponding variable values and selection status *before* the interventions;
- $\mathcal{E} = \{\epsilon_i\}_{i=1}^D$ are exogenous noise variables capturing common external influences on $X$ and $X^*$.

$\mathcal{G}^{\mathcal{I}}$ consists of the following four types of direct edges:

- Direct causal effect edges in both worlds: $\{X_i \to X_j, X_i^* \to X_j^*\}_{i \to j \in \mathcal{G}, \ i,j \in [D]}$;
- Selection edges in the counterfactual basal world: $\{X_i^* \to S_j^*\}_{i \to S_j \in \mathcal{G}, \ i \in [D], \ j \in [T]}$;
- Exogenous influence edges: $\{\epsilon_i \to X_i, \epsilon_i \to X_i^*\}_{i \in [D]}$;
- Edges representing mechanism changes due to interventions: $\{\zeta_k \to X_i\}_{i \in I^{(k)}, k \in [K]}$.

Using structural equation model (SEM) notation, denote by $f_i^*$ the generating function that maps $(X_{\text{pa}_{\mathcal{G}}(i)}^*, \epsilon_i)$ to $X_i^*$, and $\{f_i^{(k)}\}_{k=0}^K$ the functions (if exists) that maps $(X_{\text{pa}_{\mathcal{G}}(i)}, \epsilon_i)$ to $X_i$ in corresponding pure observational or interventional distributions (where $\zeta = \mathbb{1}_k$). We assume the counterfactual basal world operates the same as the pure observational world. For each intervention targeted at $I^{(k)}$, we also assume non-targeted variables' generating functions are invariant to this intervention:

$$\forall k \in \{0\} \cup [K], \quad \forall i \in [D] \backslash I^{(k)}, \quad f_i^{(k)} \text{ exists and } f_i^{(k)} \equiv f_i^*. \tag{C.1}$$

The graphical structure $\mathcal{G}^{\mathcal{I}}$ with the invariance constraint Equation (C.1) completes our definition.

An illustrative example of the alternative one-for-all interventional twin graph is shown in Figure 7. Readers may compare it with the ones shown in Figure 3.

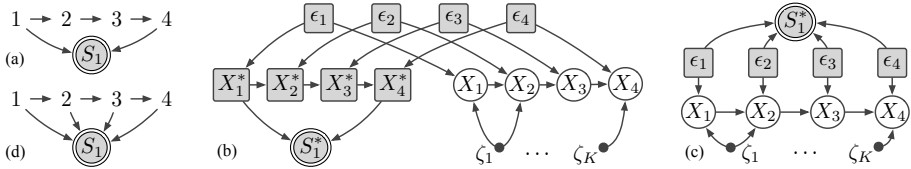

Figure 7: The *alternative one-for-all interventional twin graph* (Definition 6). The white $X$ nodes and solid $\zeta$ nodes are observed, forming the reality world, where observations or interventions are conducted. The grey nodes are unobserved, of which squares are latent variables and double circles are selection variables. The unobserved $X^*$ and $S^*$ variables form the counterfactual basal world where interventions have not been applied. The two worlds are linked by $\mathcal{E}$, latent exogenous noise variables that commonly influence $X$ and $X^*$.

Figure 8: Examples showing how d-separations in one-for-all twin graph can lose information. (a) DAG $\mathcal{G}$. (b) $\mathcal{G}^{\mathcal{I}}$ for arbitrary $\mathcal{I}$. (c) $\overline{\mathcal{G}^{\mathcal{I}}}$ constructed from (b) as in Lemma 7 where all d-separations among $X \cup \zeta$ remain the same. (d) Another DAG $\mathcal{H}$ whose $\overline{\mathcal{H}^{\mathcal{I}}}$ is also (c).

As we have shown in Lemma 2, referring to the original DAG $\mathcal{G}$ is not Markovian for interventional distributions. Then, does the one-for-all model $\mathcal{G}^{\mathcal{I}}$ fully capture the interventional distributions? The answer is still no: $\mathcal{G}^{\mathcal{I}}$ may be unfaithful, i.e., there are CIs not implied by d-separations in $\mathcal{G}^{\mathcal{I}}$. A trivial example is that, one could construct the $\mathcal{G}^{\mathcal{I}}$ for the pest control example where $\mathcal{G} = 1 \to 2 \to 3 \to S_1$ and $\mathcal{I} = \{\varnothing, \{1\}\}$. $\mathcal{G}$ encodes a d-connection $X_1 \not\perp\!\!\!\perp_d X_3 | X_2, \zeta, S^*$, which is indeed the case for $X_1 \not\perp\!\!\!\perp X_3 | X_2$ in $p^{(1)}$. However, if we let $\mathcal{I} = \{\varnothing, \{3\}\}$, the d-connection $X_1 \not\perp\!\!\!\perp_d X_3 | X_2, \zeta, S^*$ still holds, while this time, in $p^{(1)}$ it is actually $X_1 \perp\!\!\!\perp X_3 | X_2$. To explain this, one may notice that intervention on $X_3$ does not change the values of $X_2$ for individuals, and therefore in this case, when conditioning on $X_2$, the counterfactual value $X_2^*$ is automatically conditioned on, blocking the open path.

We have shown a major issue of the one-for-all twin graph, or, the issue of splitting every vertices into two worlds: the d-separations over the graph do not fully capture the conditional independencies implied. Then, one may wonder, what is exactly the d-separations in this one-for-all twin graph? We show the following lemma to characterize these d-separations, and also to illustrate why the one-for-all twin graph misses key distributional information:

**Lemma 7.** *For each one-for-all twin graph $\mathcal{G}^{\mathcal{I}}$, construct another DAG $\overline{\mathcal{G}^{\mathcal{I}}}$ by removing the $X^*$ nodes and associated edges, and adding edges $\{\epsilon_i \to S_j^*\}_{i \in \mathrm{an}_{\mathcal{G}}(S_j), \, i \in [D], \, j \in [T]}$, i.e., selection now applies ancestrally on exogenous noises. Then, $\mathcal{G}^{\mathcal{I}}$ and $\overline{\mathcal{G}^{\mathcal{I}}}$ entail the same set of d-separations over $X \cup \zeta | S^*$.*

Examples illustrating Lemma 7 are presented in Figure 8. For each one-for-all graph $\mathcal{G}^{\mathcal{I}}$, the counterfactual values $X^*$ can be further marginalized, allowing the construction of a new graph, $\overline{\mathcal{G}^{\mathcal{I}}}$. This new graph excludes $X^*$ and includes edges from $\mathcal{E}$ ancestrally pointing to $S^*$. Importantly, the d-separation patterns remain unchanged. Subfigures (c) and (d) show this construction. Then, what is the consequence of this equivalence? Consider the following example:

Let DAGs $\mathcal{G}$ and $\mathcal{H}$ shown in (a) and (b). They share the same d-separations among $X \cup \zeta$ in their respective $\mathcal{G}^{\mathcal{I}}$ and $\mathcal{H}^{\mathcal{I}}$ for arbitrary $\mathcal{I}$, as both $\mathcal{G}^{\mathcal{I}}$ and $\mathcal{H}^{\mathcal{I}}$ are equivalent to the $\overline{\mathcal{G}^{\mathcal{I}}}$ shown in (d). Then, does this imply that the $\mathcal{G}$ and $\mathcal{H}$ are indistinguishable under any $\mathcal{I}$? The answer is no. Actually, they can already be distinguished in $p^{(0)}$, where $1 \perp\!\!\!\perp_d 3 | 2, 4, S_1$ holds in $\mathcal{G}$ but not in $\mathcal{H}$.

From above, we have shown that the d-separations alone of the one-for-all interventional twin graph can fail to characterize the distributions. There are two specific losses: one is determinism, i.e., when some unaffected variable is conditioned on, its counterfactual basal variable will also be conditioned

on; the other is the loss of sparsity of selection mechanisms, i.e., it falsely represents selection as being applied ancestrally on exogenous noise terms instead of on $X^*$ variables. While we can solve these issues by defining Markov properties in a technically heavier way, we choose to use interventional twin graphs as defined in Definition 1, where the d-separations are exactly all the conditional independence implications.

## D   RELATED WORK

In this section we give a more comprehensive review of literature.

**When only pure observational data is available.**   There are constraint-based causal discovery algorithms (Spirtes et al., 2000), score-based algorithms (Chickering, 2002), and methods that utilize properties of functional forms in the underlying causal process (Shimizu et al., 2006; Hoyer et al., 2008; Zhang & Hyvärinen, 2009). The corresponding Markov equivalence characterization can be referred to (Verma & Pearl, 1991; Meek, 1995; Andersson et al., 1997; Robins et al., 2000; Friedman et al., 2000; Brown et al., 2005).

**Two kinds of interventions.**   When experimental data is available, previous literature has considered two types of interventions to model how experimental data is generated: *hard* (or *perfect*) interventions and *soft* (or *imperfect*) interventions, also known as *mechanism change*. Hard interventions destroy the dependence between targeted variables and their direct causes, either by *deterministically* fixing the target variables to specific values, or by *stochastically* setting them to values drawn from independent random variables (Pearl, 2009; Korb et al., 2004). In contrast, soft interventions do not destroy the aforementioned dependence, and they modify the functional form that characterizes the causal generating mechanism of targeted variables (Tian & Pearl, 2001; Eberhardt & Scheines, 2007).

**The earliest attempts on interventional causal discovery.**   The earliest Bayesian methods are introduced by (Cooper & Yoo, 1999; Eaton & Murphy, 2007), compute the posterior distribution of DAGs using both observational and interventional data. These methods however did not address critical challenges like identifiability or equivalence class characterization. (Tian & Pearl, 2001) is the first to consider identifiability and the Markov equivalence for interventional causal discovery. They consider the single-variable interventions with mechanisms change (soft interventions). A graphical criterion for two DAGs being indistinguishable is given, but no graphical representation for such equivalence class is characterized.

**Treatments on hard interventions.**   (Hauser & Bühlmann, 2012) first considers the characterization of MEC with hard stochastic, multiple-variable interventions. There graphical criterion is based on the mutilated DAGs as introduced in §2, and the graphical representation for equivalence class is given by $\mathcal{I}$-essential graphs. Their graph criterion actually is consistent with (Tian & Pearl, 2001)'s, though the latter focuses on single-variable interventions only. The provided algorithm GIES Hauser & Bühlmann (2015) is basically utilizing the CI relations in each experimental setting and integrate results together. Under such paradigm, methods such as (Tillman & Spirtes, 2011; Claassen & Heskes, 2010) are also developed. (Wang et al., 2017) show the consistency issue of GIES when certain faithfulness assumptions are violated, and propose the new permutation-based algorithms (Wang et al., 2017).

**For soft interventions, exploiting invariance in mechanisms.**   The basic idea of using the (in)variance of causal generating mechanisms and the asymmetries among them to identify causal relations can root back to (Hoover, 1990) with its application in economics. The invariant causal inference framework (Meinshausen et al., 2016; Rothenhäusler et al., 2015; Ghassami et al., 2017; Peters et al., 2016) is developed, though they typically require parametric assumptions such as linear models and the problem is transformed to regression analysis. Such invariances is later seen as conditional independencies between an augmented exogenous domain index variable and the other variables, which further can be related to the d-separation conditions on graphs. The interventional causal discovery can then be unified with pure observational causal discovery, by viewing domain indices as causal variables. Such representation has been discussed in e.g., (Korb et al., 2004; Newey & Powell, 2003). In the causal discovery field, it is proposed by (Zhang et al., 2015) and got formalized in (Zhang et al., 2017; Huang et al., 2020; Mooij et al., 2020), providing a unified way of

seeing data from multiple domains with mechanism change. The graphical criterion for Markov equivalence under soft interventions is given in (Yang et al., 2018). As with the earlier consistency between (Hauser & Bühlmann, 2012) and (Tian & Pearl, 2001), it is shown that as long as the pure observational data is available, the equivalence condition for hard interventions and soft interventions are the same. The issue of unknown intervention targets, also known as "fat hand" issue, is also directly solvable from the augmented graph (Squires et al., 2020; Jaber et al., 2020).

**When latent variables are involved.** In the pure observational data and nonparametric causal discovery setting, the frameworks of MAG and FCI have been well established(Richardson & Spirtes, 2002; Zhang, 2008). For interventional causal discovery, various methods have been proposed to address latent variables (Hyttinen et al., 2013b; Triantafillou & Tsamardinos, 2015; Kocaoglu et al., 2017; Eaton & Murphy, 2007; Magliacane et al., 2016). They are either lying under the umbrellas of FCI and the augmented DAG frameworks, or using parametric assumptions.

**Another parallel line of study: active experimental design.** Active experimental design is a closely related area of research, but with a distinct focus. In the interventional causal discovery setting, the interventional data are passively observed. In active experimental design however, we have control over experiments. The aim is then to select specific targets for intervention in a sequence of steps to efficiently uncover the final DAG. It can be roughly drawn into two lines. The first line is graph based methods, such as (He & Geng, 2008; Eberhardt, 2008; Eberhardt et al., 2005; 2010; Hyttinen et al., 2013a; Shanmugam et al., 2015; Kocaoglu et al., 2017; Ghassami et al., 2018), which characterize the equivalence at each step, and considers counting the DAGs in each equivalence class so as to find the next step interventions that can possibly maximally reduce the DAG search space. The second line is Bayesian based methods, such as (Tong & Koller, 2001; Murphy, 2001; Agrawal et al., 2019; Sussex et al., 2021; Tigas et al., 2022; Zhang et al., 2023), which treat the problem as an optimization problem.

**Modelling the interaction between reality and counterfactual world.** At first glance, our model seems similar to the *twin network* defined by (Balke & Pearl, 1994), as both link reality and counterfactual worlds through exogenous noise. This is where our name 'twin' is echoing. However, key differences exist. Twin networks, or single world intervention graphs (SWIGs (Richardson & Robins, 2013)), are used as diagrams to guide counterfactual queries when the structure is known, while we discover structure from data. Their CI queries are in forms of $X_A^*; X_B|X_C$, so as to find CIs to calculate counterfactual quantities from the reality quantities, while we check the (in)variances of $p(X_A|X_C)$. Also, they only model hard interventions while we handle soft ones. The most relevant model we find is from (Ribot et al., 2024) with a different focus on imputation. They assume linearity and have non-fixed $S^*$ while we explore Markov properties nonparametrically.

**Causal discovery in the presence of selection bias.** For causal discovery from selection biased data, one of the earliest approaches is the Fast Causal Inference (FCI) algorithm (Spirtes et al., 1995), which leverages conditional independence relations to infer causal structures while accounting for both latent confounders and selection bias. Following this foundational work, most subsequent methods have also relied on conditional independence constraints to recover causal relationships (Hernán et al., 2004; Tillman & Spirtes, 2011; Evans & Didelez, 2015; Versteeg et al., 2022; Zheng et al., 2024). Several parametric approaches have been developed for bivariate causal orientation (Zhang et al., 2016; Kaltenpoth & Vreeken, 2023). Other research efforts with selection bias have focused on causal inference (Bareinboim et al., 2014; Correa et al., 2019). Structure learning and inference results can follow different methodological directions depending on the problem setting. A particularly relevant area is the study of data missingness, which shares similarities with selection bias. For instance, in cases involving *self-masking* missingness, the true data distribution and model parameters may be unidentifiable, making causal inference infeasible (Mohan et al., 2013; Mohan, 2018). However, the causal structure may still be recoverable (Dai et al., 2024).

## E   SUPPLEMENTARY EXPERIMENTAL DETAILS AND RESULTS

We use the implementation of IGSP, UT-IGSP, and JCI-GSP from the `causaldag` package (Chandler Squires, 2018), and the implementation of CD-NOD from the `causal-learn` package (Zheng et al., 2023). We use the Fisher Z test to examine conditional relations. The significance level is set

to 0.05 for 5 and 10 variables, and 0.01 for 15 and 20 variables.

In what follows, we demonstrate the resulting graphs on real datasets. In these graphs, the red square nodes and their outgoing edges represent the intervention targets across different interventions. Among observed variables, $X_i \to X_j$ indicates a causal edge from $X_i$ to $X_j$ and $X_j$ is not ancestrally selected; $X_i — X_j$ indicates that both $X_i$ and $X_j$ are ancestrally selected; $X_i \circ\!\!-\!\!\circ X_j$ indicates that each endpoint may vary in the equivalence class. For simplicity in showing the DAG and the estimated intervention targets at the same time, we put multiple intervention indicators, observed variables, and the selection variables, if any, all into one graph. This is merely for presentation ease; according to Example 1 and Lemma 7, such graphs do not characterize the CI relations in data. The edge interpretation applies to all figures below.

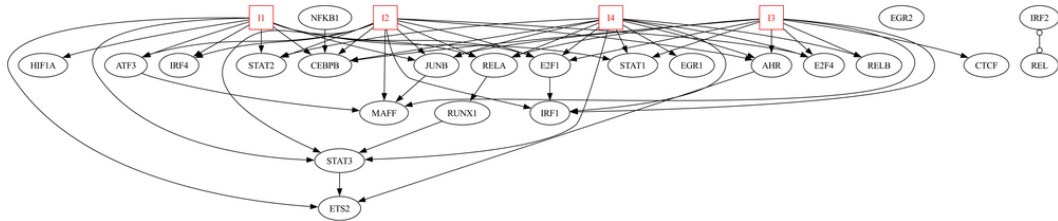

Figure 9: Causal structure estimated by our CDIS method on a single-cell perturbation data, i.e., sciPlex2 (Peidli et al., 2024). We discovered some validated regulatory relationships like *RELA →
RUNX1* and *JUNB → MAFF*, since RELA has been implicated as a *RUNX3* transcription regulator (Zhou et al., 2015) and *Maf* family was upregulated by *JunB* (Koizumi et al., 2018), despite the extensive co-regulation between these two genes (Kataoka et al., 1994). Besides, the link between *ETS2* or *RUNX1* and *STAT3* may encounter the risk of spurious correlation induced by selection (i.e., conditioning on a particular cell line). It is supported by previous studies since it is reported Runx1 and Stat3 synergistically driving stem cell development in epithelial tissues trough Runx1/Stat3 signalling network (Scheitz et al., 2012; Sarper et al., 2018), while TF Ets2 together with p-STAT3 activation induce cathepsins K and B expression in human rheumatoid arthritis synovial fibroblasts (RASFs) (Singh et al., 2021).

| Category | Variable | Meaning |
|---|---|---|
| Main outcome | GPA_year1 | 1st year GPA |
| | goodstandi~1 | Good standing in year 1 |
| | prob_year1 | On probation in year 1 |
| | credits_ea~1 | Credits earned in year 1 |
| | mathsci | Number of math and science credits attempted |
| Personal backgrounds | female | Sex (Female dummy) |
| | age | Age |
| | english | Whether Mother tongue is English |
| | gpa0 | High school GPA |
| Other covariates | hcom | Whether Lives at Home |
| | chooseUTM | Whether At first choice school |
| | work1 | Whether Plans to work while in school |
| | dad_edn | Father education |
| | mom_edn | Mother education |
| | lm_rarely | Whether Rarely puts off studying for tests |
| | lm_never | Whether Never puts off studying for tests |
| | lastmin | how often do you leave studying until the last minute for tests and exams |
| | graddeg | Whether Wants more than a BA |
| | finish4 | Whether Intends to finish in 4 years |

Table 1: Description of variables in the educational dataset

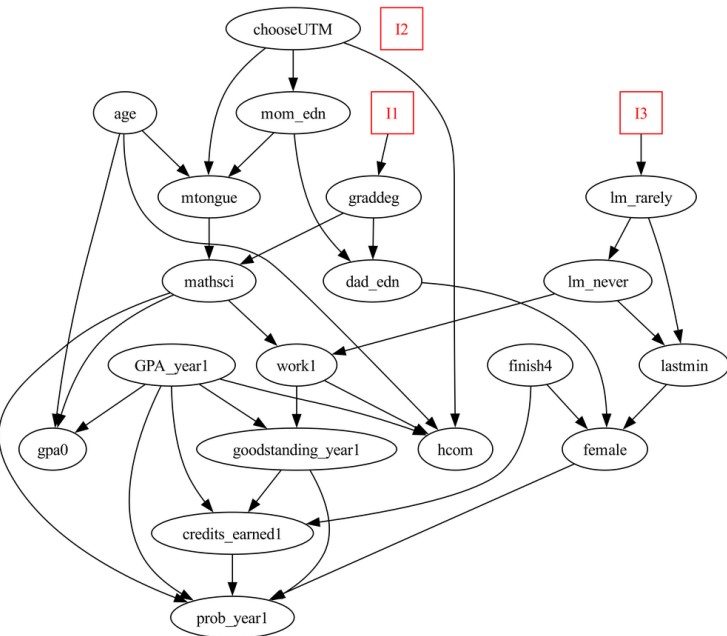

Figure 10: Causal structure estimated by our CDIS method on an educational dataset over all students. The interventional indicators I1 represents SSP, I2 represents SFP, and I3 represents SFSP.

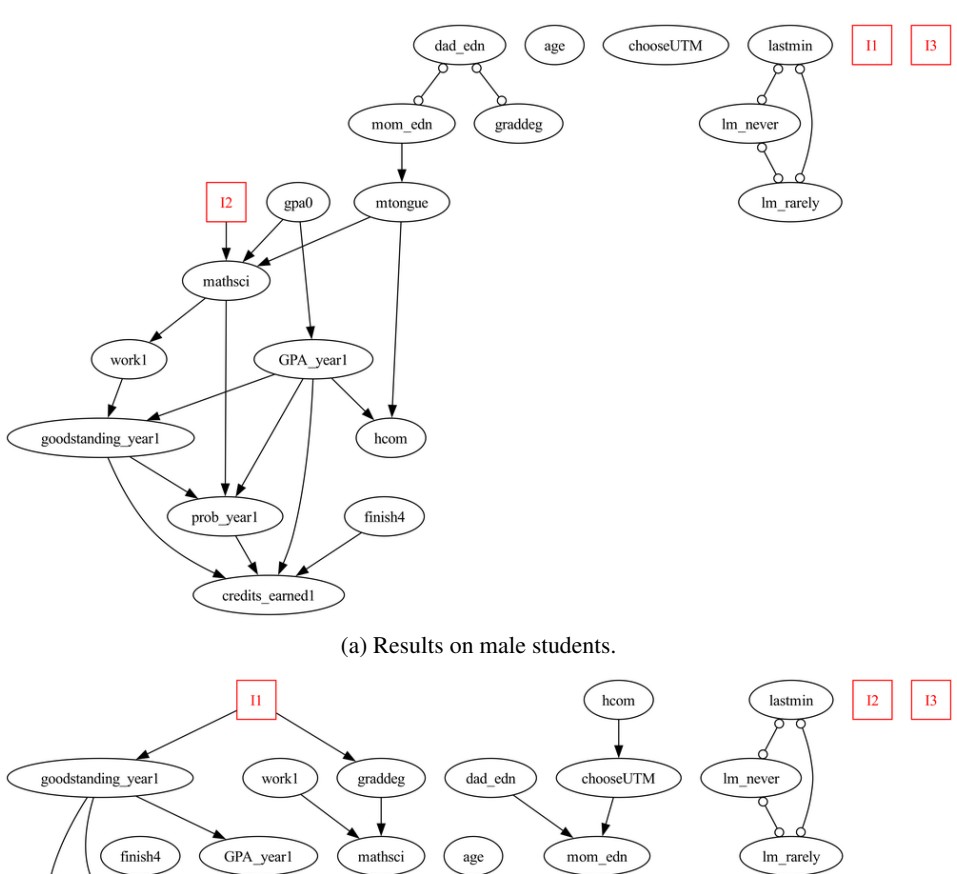

(a) Results on male students.

(b) Results on female students.

Figure 11: Causal structure estimated by our CDIS method on an educational dataset conducted with subgroup analysis stratified by gender.

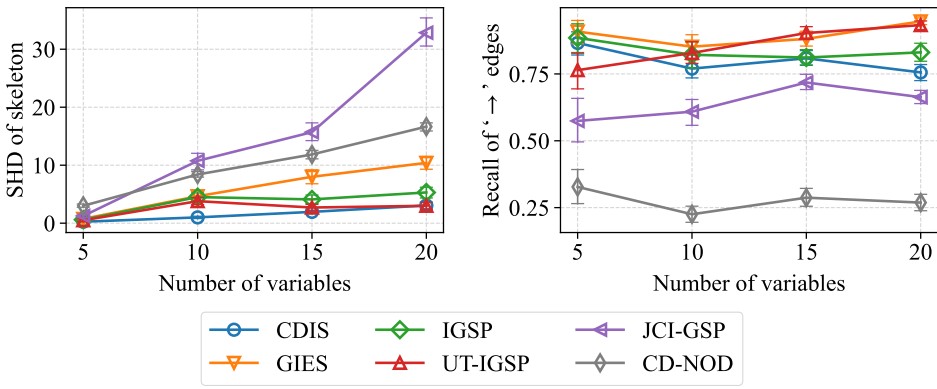

Figure 12: Empirical results across different numbers of variables. The two figures show the SHD of the PAG skeletons, and the recall of the '→' edges in PAGs. The error bars illustrate the standard errors from 20 random simulations.

