# OpenReview forum: "When Selection Meets Intervention: Additional Complexities in Causal Discovery"
_ICLR.cc/2025/Conference — ICLR 2025 Oral_

### Official Review · Reviewer_9cf1 · 2024-11-02

**Soundness:** 3
**Presentation:** 2
**Contribution:** 3
**Rating:** 8
**Confidence:** 3

**Summary:**

This paper addresses the important but overlooked problem of selection bias in interventional causal discovery, where subjects are selectively enrolled in experiments (e.g., drug trials focusing on patients with specific conditions). The authors show that existing methods for interventional causal discovery fail to properly handle selection bias, as subtle differences in when and where interventions occur can lead to inconsistent conditional indepenence relations. To address this, they introduce a novel graphical model called the "interventional twin graph" that explicitly accounts for both the observed world (where interventions are applied) and the counterfactual world (where selection occurs before interventions), along with characterizing its Markov properties and equivalence classes. They develop a provably sound algorithm called CDIS (Causal Discovery from Interventional data under potential Selection bias) that can identify causal relations and selection mechanisms from data with soft interventions and unknown targets. Through experiments on both synthetic data and real-world applications in biology and education, they demonstrate that their method achieves higher precision in identifying true causal relations compared to existing approaches when selection bias is present.

**Strengths:**

- This paper studies a relevant and interesting problem - both mathematically and philosophically. It considers the question: "What does selection bias actually mean" and proposes a sound answer and the necessary mathematical framework to deal with such situations.
- Based on the framework to treat selection bias, a sound and complete causal discovery algorithm is proposed.
- The method is evaluated not only on synthetic data, but on real-world examples as well. This provides some confidence that it may be useful in practical applications.

**Weaknesses:**

The biggest weakness I see is the presentation of the paper. The first two sections are dense, but give a good introduction and motivation to the problem, based on good illustrations in Examples 1 and 2.

However, Section 3 is the most painful piece of text I have read in a while. It relies mostly on mathematical notation to bring across the main points and lacks the contextualization in prose. I appreciate that examples are given in Section 3, but even those are a bit cryptic and fail to provide an accessible intuitive understanding. I suppose there are three main reasons for this: (1) Writing about complex SCMs is inherently difficult and a certain level of formalism is necessary - not much you can do here. (2) The amount of content in the main paper, given the page limit might a bit too much. Some of the more technical parts could be relegated to the appendix and exchanged for more contextualization. (3) The text could consider the reader's state of mind more. Some examples:

- L211f: introducing the functions $f^*$ uses the mathematical symbols for the corresponding variables in the counterfactual basal world to introduce them, but does not use the word "counterfactual". That means as a reader, I either have it in working memory, or I have to go back to the definition and jump back again to the sentence to parse it.
- As far as I can tell, abbreviations like "CI" and "MAG" aren't defined, or used before they are defined, e.g. "PAG".

Such presentation choices add unnecessary mental effort for understanding, and I would think twice if I'd go back to this paper and build on it for future work - not because it's wrong, but because of the mental effort to access the information.

**Questions:**

-

---

> ### Author Response · Authors · 2024-11-22
> **Response to Reviewer 9cf1 (Part 1/2)**
>
> We appreciate the reviewer's constructive comments and helpful feedback. Please see below for our response.
>
> ---
>
> **(Q1)**  The reviewer identifies the biggest weakness as presentation in our original submission.
>
> **R:** We are grateful for the reviewer's comments. In response to your feedback, we have carefully rewritten our theoretical sections in the updated manuscript. While the technical results remain unchanged, we have significantly improved the way we deliver these results.
>
> Key improvements are summarized below:
>
>  - **Optimal model first:** We directly give the optimal model, deferring its detailed development to the appendix, to improve readability.
> 	 - In our original submission, we presented our theory development sequentially: starting with an intuitive model, then demonstrating its flaws, and finally making corrections to reach the optimal model. While this approach was helpful for understanding the technical progression, it may have impacted readability.
>  - **Graph examples and explanations:** We provide each definition and theorem with graph examples and explanations afterwards, so that readers can try to apply the graphical criteria and understand them more easily.
>  - **Context for lemmas:** We introduce each lemma alongside the specific question it aims to answer, for logical clarity.
>  - **Running examples:** We use two running examples—the clinical and the pest control one—throughout, to illustrate all main results.
>  - **Concepts introduction:** We introduce all technical concepts, such as maximal ancestral graphs (MAGs), before they are used.
>
> We appreciate the reviewer's time devoted. We hope you find the updated manuscript less of a painful read this time around.

---

> ### Author Response · Authors · 2024-11-22
> **Response to Reviewer 9cf1 (Part 2/2)**
>
> We are encouraged that the reviewer recognizes our work as "studying a relevant and interesting problem - both mathematically and philosophically".
>
> We would like to highlight that our contribution goes beyond solving a specific problem; **we identify a new problem setting and establish a general framework to understand this setting, namely, interventional studies under selection bias.**
>
> In real-world scenarios, selection indeed always meets intervention, as real experiments are typically designed with specific scopes and objectives (e.g., participants in a drug trial are usually patients of the relevant disease).
>
> To this end, we hope this work provides **a foundation for various potential practical extensions**, such as incorporating latent variables for causal discovery and deriving causal inference results.
>
> **We want to thank the reviewer again for all the valuable feedback.**

---

> > ### Comment · Reviewer_9cf1 · 2024-11-26
> >
> > I thank the authors for their detailed rebuttal and changes made to the manuscript to aid readability. I have adapted my score accordingly.

---

> ### Author Response · Authors · 2024-11-27
>
> We sincerely thank the reviewer for your recgnition and your valuable feedback that helps improving the presentation of our submission.

---

### Official Review · Reviewer_kjX2 · 2024-11-03

**Soundness:** 3
**Presentation:** 3
**Contribution:** 3
**Rating:** 8
**Confidence:** 2

**Summary:**

This paper considers the problem of interventional causal discovery in the context of selection bias. The paper first explains why simply augmenting the causal DAGs with a selection variable is insufficient. The paper then goes on to introduce the concept of a twin graph, in which every node is replicated in a counterfactual/"basal" world, and there is a defined set of rules by which the relationships of the basal world are constructed. Based on this construction, the paper goes on to define the set of d-separations that are implied by this model and establish the graphical criteria for Markov equivalence. The paper provides an algorithm to recover the Markov equivalence class and evaluates it on both simulated and real-world data.

**Strengths:**

The paper clearly lays out the problem of selection bias in causal discovery and why certain natural approaches to the problem are not sufficient. The paper also puts forward a very general solution to the problem and considers its consequences. Overall the paper is well-written, despite being notation-heavy.

**Weaknesses:**

One thing that was unclear to me is how complete is the paper's characterization of Markov equivalence classes in the given model. I would think that the Markov equivalence class would encode all DAGs with the same conditional independence structure (i.e. the right-hand side of the implications in Theorem 1). However, the equivalence structure is defined with respect to the left-hand side of the implications in Theorem 1. This would seem to imply that the equivalency classes are not as fine-grained as they potentially could be.

The algorithm provided also suffers from this issue, in that the authors point out that it may not be complete. It is not clear to me how useful it is to have a causal discovery algorithm that is sound but not complete. The trivial algorithm that says there are no causal relationships is sound but not useful.

The simulation study not reporting any information on completeness is disheartening. While I understand that the paper does not contain any guarantees on completeness, in the simulations there is access to the ground truth. So it is hard to see how there is no way to evaluate the ability to discover some fraction of those relationships.

At a higher level, I'm not sure how much of the framing of the paper is specific to the selection problem. It seems like the approach of the paper is tackling the more general problem of causal discovery with unobserved latent variables. If that is not the case, then the paper should explain how their methods do not generalize to the latent variable setting.

**Questions:**

Can you think of any ways in which you could evaluate the completeness of your proposed algorithm in the simulations? In a causal discovery paper, we should be concerned with making true discoveries, not just avoiding false discoveries.

Does the paper's approach generalize to arbitrary latent variables?

---

> ### Author Response · Authors · 2024-11-22
> **Response to Reviewer kjX2 (Part 1/4)**
>
> We sincerely appreciate the reviewer's constructive comments and helpful feedback. Please see below for our response.
>
> ---
> **(Q1)**  The reviewer wonders if our characterization of Markov equivalence class is as fine-grained as possible.
>
> **A:** We thank the reviewer for this thoughtful question, and would like to note that **we have indeed characterized the Markov equivalence class to the finest-grained level possible,** capturing all conditional independence (CI) implications. Specifically:
>  - Our Theorem 1 (Markov properties) is expressed in the following form:
>    > "For any graph $\mathcal{G}$ and any distribution $p$ generated from $\mathcal{G}$, **if** [some graphical criteria holds in $\mathcal{G}$], **then** [some conditional independence holds in $p$]."
>
>    The reason why we do not use the "if and only if" statement is just because faithfulness is not assumed at this stage—not due to any missing conditional independence implications.
>
>  - In essence, the left-hand side (graphical criteria) and right-hand side (conditional independence implications) characterize the same fine-grained level of equivalence. To view this, note that Theorem 1 can equivalently be expressed as:
>    > "For any graph $\mathcal{G}$, [some graphical criteria holds in $\mathcal{G}$], **if and only if** [some conditional independence holds in all distributions generated by $\mathcal{G}$]."
>
> We thank the reviewer again for highlighting this, and hope the above explanation clarifies that our equivalence classes are indeed fully characterized.
>
>
> ---
> **(Q2)**  The reviewer wonders if our approach generalizes to arbitrary latent variables.
>
> **A:** We appreciate this insightful question. **The short answer is, yes.** Specifically,
>    - Similar to how the augmented DAG framework offers a unified way to understand interventional data without selection bias, our interventional twin graph framework **offers a unified way to understand** interventional data with selection bias.
>    - Then, various extensions can be incorporated into this framework, just as they are in the augmented DAG paradigm (discussed in Section 2.1 in our manuscript). **Latent variables is one of such extensions.** Others include e.g., deriving causal inference results.
>    - For latent variables, **the technical route is straightforward**: one may modify the specific graphical criteria of constructing maximal ancestral graphs (MAGs) (Lemmas 4 to 6 in our updated manuscript) and adjust the orientation rules and edge interpretations in Algorithm 1. While details may differ, the core concept remains unchanged.
>    - In this work, we focus solely on selection bias, as we aim to thoroughly characterize its own complexities, highlight its impact on interventional studies, and maintain clarity in our presentation.
>    - One philosophically interesting observation, as discussed in Section 2.3, is that **latent variables are easier to incorporate than selection bias for interventional studies.** This is because, when selection bias is absent, the two forms of intervention (on existing individuals vs. from scratch) yield the same interventional data at the distribution level, even with latent variables. Selection bias, however, behaves quite differently.

---

> ### Author Response · Authors · 2024-11-22
> **Response to Reviewer kjX2 (Part 2/4)**
>
> **(Q3)**  The reviewer raises concerns about the completeness guarantee of our algorithm, CDIS.
>
> **A:** We appreciate the reviewer for these constructive comments. In what follows, we address: (1) to which extent CDIS is complete, (2) how and why proving full completeness is technically challenging, (3) how a sound but not fully complete algorithm can also be useful, and (4) completeness evaluation from simulation experiments.
>
> 0. **Before proceeding, we introduce the following terms** (for more details, please kindly refer to Section 3.3 and Appendix. A of our updated manuscript):
> 	  - Maximal ancestral graph (**MAG**): a graph to represent conditional independence relations among observed variables in the presence of latent variables and selection bias.
> 	  - Partial ancestral graph (**PAG**): a graph to represent an equivalence class of MAGs, just like how a CPDAG represents an equivalence class of DAGs.
> 	  - Fast causal inference (**FCI**) algorithm: an algorithm that recursively applies the orientation rules to a PAG until no further orientations can be made. This is similar to how the PC algorithm uses Meek rules to orient edges beyond v-structures.
>
> 1. **To which extent CDIS is complete.**
>    - **We conjecture that CDIS is fully complete,** that is, it produces the maximally informative output about the Markov equivalence class (defined in Theorem 2). This conjecture is supported by brutal search on \~50,000 (DAG, targets) pairs with up to 15 vertices. For the PAG that CDIS outputs on each pair, we enumerate all the MAGs consistent with it, and verify if there exist targets so that they are Markov equivalent to the original pair. So far we found no counterexamples.
>    - **CDIS is guaranteed to be complete on pure observational data,** since it is built on the result that FCI gives on pure observational data, which is guaranteed to be maximal informative with respect to pure observational data. Furthermore, CDIS incorporates additional information from interventional data, making its result even more informative than that.
>
> 2. **How and why proving full completeness is technically challenging.**
>    - **This challenge is also reflected in the completeness establishment of FCI itself.** Proving completeness of orientation rules can be challenging, as it requires showing that all the possible graphical patterns are enumerated. For instance, though FCI has been widely adopted since its debut in 1999 [1], its completeness result (without background knowledge) was not established until 2005 (partially [2]) and 2008 (fully [3]).
>    - **Our algorithm, CDIS, can be viewed as FCI with background knowledge.** To determine edges on the original PAG from PAGs obtained from interventional data, the refining step in CDIS relies on our graphical patterns on interventional twin graphs (Lemma 6) as background knowledge.
>    - **Why FCI with background knowledge is not complete.** For illustration, consider the analogous case of the PC algorithm with Meek rules, which orient more direct edges starting from those of v-structures. Without background knowledge, Meek's R1-R3 rules are complete, i.e., can reach the correct CPDAG. However, adding background knowledge requires an additional R4 rule to ensure completeness [4]. Similarly, FCI's current orientation rules (R1–R10) are analogous to Meek's R1–R3, but in general, the counterpart of Meek's R4 in FCI's context remains unknown [5]. Consequently, the completeness of FCI with general background knowledge has not yet been established.
>
> 3. **How a sound but not fully complete algorithm can also be useful.**
>    - While completeness is desirable, sound algorithms can still be useful, if it can reliably output informative results, especially when proving completeness is difficult or complete algorithms in practice add more complexities than benefits. For instance,
>    - The FCI algorithm itself has been widely adopted to deal with hidden confounders before its completeness was established (1999 to 2008).
>    - Even now FCI is complete with respect to the Markov equivalence class, it is not complete with respect to the distribution equivalence class, due to additional (in)equality constraints given by latent variables [6].
>    - For cyclic causal discovery, the CCD algorithm [7] has been widely adopted but it is incomplete, since with cycles, a d-separation may not imply a conditional independence.
>    - Overall, these examples illustrate that an algorithm does not have to be fully complete to be useful; its value depends on **how close it is on the spectrum towards completeness—both theoretically and empirically.**
>    - Therefore, we show our empirical results on simulations below (note that our position on the theoretical axis is already shown in above point 1).

---

> ### Author Response · Authors · 2024-11-22
> **Response to Reviewer kjX2 (Part 3/4)**
>
> 4. **Completeness evaluation from simulation experiments.**
>     - We especially thank the reviewer for pointing out that completeness can be evaluated with access to ground truth. Although not all directed edges in the truth DAG are identifiable, the equivalence class can at least be brutally searched.
>     - We conducted experiments on varying number of nodes with 1,000 samples, with all other configurations the same as Section 5.1. We report the F1 score, precision, and recall of the direct edges in the equivalence class. The results are provided in tables below, as well as Figure 12 in the updated manuscript.
>
>         | #Nodes   |         10          |             |             |         15          |             |             |         20          |             |             |
>         |----------|----------------------|-------------|-------------|--------------------|-------------|-------------|---------------------|-------------|-------------|
>         |   **Method**       | **F1**               | **Precision** | **Recall**   | **F1**            | **Precision** | **Recall**   | **F1**              | **Precision** | **Recall**   |
>         | **CDIS** | **0.8±0.04**         | **0.79±0.06** | **0.82±0.04** | **0.77±0.04**     | **0.71±0.05** | 0.86±0.05   | **0.75±0.05**       | **0.74±0.09** | 0.79±0.06   |
>         | IGSP     | 0.63±0.12           | 0.56±0.12    | 0.74±0.10   | 0.64±0.03         | 0.5±0.03    | 0.88±0.04   | 0.5±0.05           | 0.37±0.05    | 0.76±0.05   |
>         | UT-IGSP  | 0.71±0.05           | 0.64±0.07    | 0.8±0.03    | 0.68±0.05         | 0.54±0.04   | **0.9±0.05** | 0.56±0.07          | 0.42±0.07    | **0.83±0.03** |
>         | JCI-GSP  | 0.32±0.06           | 0.24±0.05    | 0.5±0.08    | 0.4±0.05          | 0.28±0.04   | 0.7±0.07    | 0.31±0.04          | 0.2±0.03     | 0.68±0.04   |
>
> 	 - We see that indeed, our algorithm empirically achieves both high precision and recall, supporting its guaranteed soundness and conjectured completeness.

---

> ### Author Response · Authors · 2024-11-22
> **Response to Reviewer kjX2 (Part 4/4)**
>
> We are encouraged that the reviewer recognizes our work as "putting forward a very general solution to the problem".
>
> We would like to highlight that our contribution goes beyond solving a specific problem; **we identify a new problem setting and establish a general framework to understand this setting, namely, interventional studies under selection bias.**
>
> In real-world scenarios, selection indeed always meets intervention, as real experiments are typically designed with specific scopes and objectives (e.g., participants in a drug trial are usually patients of the relevant disease).
>
> To this end, we hope this work provides **a foundation for various potential practical extensions**, such as incorporating latent variables for causal discovery and deriving causal inference results.
>
> **We want to thank the reviewer again for all the valuable feedback.**
>
>
> ---
>
> [1] Spirtes, Peter L., Christopher Meek, and Thomas S. Richardson. "Causal inference in the presence of latent variables and selection bias." (1999).
>
> [2] Ali, Ayesha R., et al. "Towards characterizing Markov equivalence classes for directed acyclic graphs with latent variables." (2005).
>
> [3] Zhang, Jiji. "On the completeness of orientation rules for causal discovery in the presence of latent confounders and selection bias." (2008).
>
> [4] Meek, Christopher. "Causal inference and causal explanation with background knowledge." (1999).
>
> [5] Andrews, Bryan, Peter Spirtes, and Gregory F. Cooper. "On the completeness of causal discovery in the presence of latent confounding with tiered background knowledge." (2020).
>
> [6] Pearl, Judea. "On the testability of causal models with latent and instrumental variables." (1995).
>
> [7] Richardson, Thomas S. "Discovering cyclic causal structure." (1996).

---

> > ### Comment · Reviewer_kjX2 · 2024-11-26
> >
> > Thank you to the authors for your thorough response. All my concerns have been addressed, and I have updated my score.

---

> ### Author Response · Authors · 2024-11-27
>
> We are delighted to know that our response well addressed your concerns. Thank you once again for your constructive comments about completeness.

---

### Official Review · Reviewer_5bYr · 2024-11-03

**Soundness:** 3
**Presentation:** 3
**Contribution:** 3
**Rating:** 8
**Confidence:** 3

**Summary:**

This paper addresses the limitations of current graphical models and causal discovery methods in handling pre-intervention selection bias in interventional causal discovery. To overcome these limitations, the authors propose a novel twin graph model that effectively captures both the observed world and the counterfactual world where selection occurs. The paper establishes the Markov properties of this new model and introduces a provably sound algorithm for identifying causal relationships. The effectiveness of this approach is demonstrated through synthetic and real-world experiments.

**Strengths:**

1. The problem of selection bias is important yet often overlooked in existing interventional causal discovery. The setting is general.
2. This paper is technically sound, with clear formulation of the causal DAG and Markov properties.
3. The illustrative examples enhance the paper's clarity, helping readers better understand the concepts.

**Weaknesses:**

More comprehensive results and explanations of the empirical studies would be beneficial to support the effectiveness of the proposed algorithm. For example:
1. For the simulation, can you report the proportion of true causal edges estimated as directed edges by the algorithm? Additionally, a comparison of the output graphs with the ground truth would illustrate how the new algorithm perform differently from other methods under selection bias.
2. For the gene application, can you provide more comprehensive analysis of the result?
3. For the education dataset, can you explain why the pre-intervention selection bias is a potential issue? Highlighting and interpreting key information of the resulting graphs would be helpful, as the current graph and variable names are difficult to follow.

Minor comments about clarity:
- The notation in this setting is dense and improvements of readability can be helpful for readers less familiar with the area. For example, "CI" in line 93 and different types of arrows in Example 7 can be clarified before their first appearance.
- Typos: line 48 "existing", line 295 "false".

**Questions:**

1. What are the main challenges that affect the precision and completeness of the algorithm? How sensitive is it to the unmeasured confounders?
2. If selection bias is detected through diagnostics, how can this information be leveraged to help causal discovery?
3. Any data points on the computational cost of the algorithm?

---

> ### Author Response · Authors · 2024-11-22
> **Response to Reviewer 5bYr (Part 1/4)**
>
> We sincerely appreciate the reviewer's encouragement and insightful feedback. Please see below for our response.
>
> ---
> **(Q1)**  The reviewer wonders the main challenges that affect the precision and completeness of the algorithm.
>
> **A:**  Thank you for the question. We summarize the challenges as follows:
>
> + **Empirical challenges:** **Conditional independence (CI) tests** may not always be accurate, especially with insufficient samples. This leads to potential error propagation in orientation rules. More robust empirical methods to handle either CI tests, or conflicts in orientations, are to be developed.
> + **Theoretical Challenges:**
> 	+ **Unmeasured confounders.** We list it here merely because in this specific work we assume causal sufficiency. However, **it is not really challenging.** Specifically,
> 		+ Similar to how the augmented DAG framework offers a unified way to understand interventional data without selection bias, our interventional twin graph framework **offers a unified way to understand interventional data with selection bias.**
>         + Then, various extensions can be incorporated into this framework, just as they are in the augmented DAG paradigm (discussed in Section 2.1 in our manuscript). **Latent variables is one of such extensions.**
>         + For latent variables, **the technical route is straightforward:** one may modify the specific graphical criteria of constructing maximal ancestral graphs (MAGs) (Lemmas 4 to 6 in our updated manuscript) and adjust the orientation rules and edge interpretations in Algorithm 1. While details may differ, the core concept remains unchanged.
>         + In this work, we focus solely on selection bias, as we aim to thoroughly characterize its own complexities, highlight its impact on interventional studies, and maintain clarity in our presentation.
>          + One philosophically interesting observation, as discussed in Section 2.3, is that **latent variables are easier to incorporate than selection bias for interventional studies.** This is because, when selection bias is absent, the two forms of intervention (on existing individuals vs. from scratch) yield the same interventional data at the distribution level, even with latent variables. Selection bias, however, behaves quite differently.
> 	+ **Completeness guarantee.** Our proposed algorithm, CDIS, is provably sound. However, as CDIS can be viewed as FCI with background knowledge, no completeness result (i.e., whether it identifies all the invariant relations in the equivalence class) currently exists, and proving it is technically challenging, as is discussed after Theorem 3 in the updated manuscript. This is not a significant concern though, as:
>          + **We conjecture that CDIS is fully complete,** that is, it produces the maximally informative output about the Markov equivalence class (defined in Theorem 2). This conjecture is supported by brutal search on \~50,000 (DAG, targets) pairs with up to 15 vertices. For the PAG that CDIS outputs on each pair, we enumerate all the MAGs consistent with it, and verify if there exist targets so that they are Markov equivalent to the original pair. So far we found no counterexamples.
>          + **CDIS is guaranteed to be complete on pure observational data,** since it is built on the result that FCI gives on pure observational data, which is guaranteed to be maximal informative with respect to pure observational data. Furthermore, CDIS incorporates additional information from interventional data, making its result even more informative than that.
>          + **Evaluation from simulation experiments** suggests that our algorithm empirically achieves both high precision and recall. The detailed results are shown in our response to your question **(Q4)** below.
>
>
> Despite these challenges, we would like to note that they are common to many causal discovery algorithms. Our method provides a general enough framework to understand interventional data under selection. Experiments also show it effectively identifies true causal and selection relations.
>
>
> ---
> **(Q2)**  The reviewer wonders how the selection bias detected through diagnostics can aid causal discovery.
>
> **A:** We appreciate this constructive question. **The answer is, yes**—it can be easily integrated into our algorithm, CDIS, as background knowledge. Like other types of background knowledge (e.g., direct or ancestral causal relations), it can be input into CDIS and aligns seamlessly with our design. This is because CDIS functions exactly as FCI with background knowledge (where our graphical criteria for interventional twin graphs helps in the refining step).
>
> As you see, CDIS can be improved with additional constraints, prior knowledge, or extensions to allowing for hidden confounders. We do not focus on those variants of CDIS in this work, since our major contribution here lies in fully characterizing the data under selection bias itself.

---

> ### Author Response · Authors · 2024-11-22
> **Response to Reviewer 5bYr (Part 2/4)**
>
> **(Q3)**  The reviewer wonders the computational cost of the algorithm.
>
>
> **A:**  Thank you for this insightful question. The computational cost of our algorithm, CDIS, is similar to standard constraint-based algorithms like PC and FCI, as it mainly involves conducting conditional independence (CI) tests **among observed variables to estimate adjacencies**. This has been mentioned in the updated manuscript. Specifically,
>  - **(Q3.1)** What is the exact time complexity?
>
>    **A:** Like PC, our algorithm has a worst-case complexity exponential in the number of observed variables (the $D$ in our manuscript). However, if the underlying graph is sparse—a common and reasonable assumption—the runtime becomes polynomial [1].
>
>  - **(Q3.2)** But the interventional twin graph is more complex than a simple DAG and involves additional nodes?
>
>    **A:** This is because the true data generating mechanism is indeed more complex than and cannot reduce to a simple augmented DAG, as shown in Section 2 (Motivations). Regarding complexity of the discovery procedure, note that the above mentioned complexity of the twin graph model only impacts **how we interpret** the CI relations from data, **not the number of CI tests required,** as we can only test observed variables.
>
>  - For empirical evaluation, we have also compared the runtime (in seconds) of our algorithm with others (even though they are theoretically incorrect under selection). Results are shown as follows:
>
> 	|      Method         | 10 nodes | 15 nodes | 20 nodes|
> 	|:--------------|:---------|:---------|:---------|
> 	| **CDIS**          | 0.4±0.24 | 1.0±0.32 | 2.0±0.0  |
> 	| IGSP          | 0.6±0.24 | 0.8±0.2  | 1.2±0.2  |
> 	| UT-IGSP       | 0.6±0.24 | 0.6±0.24 | 1.4±0.24 |
> 	| JCI-GSP       | 0.4±0.24 | 0.8±0.2  | 1.4±0.24 |
>
>      We observe that these algorithms indeed operate on a similar timescale.
>
>
>
> ---
> **(Q4)**  The reviewer asks for more details about the simulation experiments. Specifically,
>
> + **(Q4.1)** "Can you report the proportion of true causal edges estimated as directed edges by the algorithm?"
>
>   **A:** Thank to your suggestion, we conducted experiments on varying number of nodes with 1,000 samples, with all other configurations the same as Section 5.1. We report the F1 score, precision, and recall with respect to the true direct edges in the equivalence class. The results are provided in tables below, as well as Figure 12 in the updated manuscript.
>
>
>
>   | #Nodes   |         10          |             |             |         15          |             |             |         20          |             |             |
>   |----------|----------------------|-------------|-------------|--------------------|-------------|-------------|---------------------|-------------|-------------|
>   |   **Method**       | **F1**               | **Precision** | **Recall**   | **F1**            | **Precision** | **Recall**   | **F1**              | **Precision** | **Recall**   |
>   | **CDIS** | **0.8±0.04**         | **0.79±0.06** | **0.82±0.04** | **0.77±0.04**     | **0.71±0.05** | 0.86±0.05   | **0.75±0.05**       | **0.74±0.09** | 0.79±0.06   |
>   | IGSP     | 0.63±0.12           | 0.56±0.12    | 0.74±0.10   | 0.64±0.03         | 0.5±0.03    | 0.88±0.04   | 0.5±0.05           | 0.37±0.05    | 0.76±0.05   |
>   | UT-IGSP  | 0.71±0.05           | 0.64±0.07    | 0.8±0.03    | 0.68±0.05         | 0.54±0.04   | **0.9±0.05** | 0.56±0.07          | 0.42±0.07    | **0.83±0.03** |
>   | JCI-GSP  | 0.32±0.06           | 0.24±0.05    | 0.5±0.08    | 0.4±0.05          | 0.28±0.04   | 0.7±0.07    | 0.31±0.04          | 0.2±0.03     | 0.68±0.04   |
>
>
>
> 	We see that indeed, our algorithm empirically achieves both high precision and recall, verifying its guaranteed soundness and conjectured completeness.
>
> + **(Q4.2)** "A comparison of the output graphs with the ground truth."
>
>   **A:**  We are especially grateful for this valuable comment. Example graphs indeed help illustrate the complexity of equivalence classes under selection, and clearly shows how CDIS differs from other algorithms.
>
>   We provide these comparisons on example graphs in Figure 14 in the updated manuscript. We see that CDIS outputs are sparser and closer to the ground truth, as it avoids misinterpreting spurious dependencies from selection bias as causation.

---

> ### Author Response · Authors · 2024-11-22
> **Response to Reviewer 5bYr (Part 3/4)**
>
> **(Q5)**  The reviewer asks for more elaboration on the real-world experiments. Specifically,
>
> + **(Q5.1)** "For the gene application, can you provide more comprehensive analysis of the result?"
>
>   **A:** Thank you for your valuable comments. We evaluate the gene regulation networks (GRNs) of 24 previous reported essential regulatory genes encoding different transcription factors (TFs) using a single-cell perturbation data, i.e., sciPlex2.
> 	+ In the sciPlex2 dataset, the A549 cells, a human lung adenocarcinoma cell line, are either exposed to one of the four different transcription modulators, including dexamethasone, nutlin-3a, BMS-345541, and suberoylanilide hydroxamic acid (SAHA), or simply treated with dimethylsulfoxide vehicle as control.
> 	+ We discoved some regulatory relationships like  _RELA_  $\rightarrow$  _RUNX1_ and  _JUNB_  $\rightarrow$  _MAFF_, which has been validated by some previous studies.
> 	+ For example, TF RELA has been previously implicated as a RUNX3 transcription regulator, and  _Maf_  family was upregulated by  _JunB_, despite the extensive co-regulation between these two genes.
> 	+ Besides, our algorithm suggests the link between transcription factor ETS2 or RUNX1 and the activation of gene  _STAT3_  may possibly reflect the true gene regulation process but also encounter the possibility of selection bias introduced when conditioning on a particular disease cell line.
> 	+ The results are supported by previous studies since it is reported Runx1 and Stat3 synergistically driving stem cell development in epithelial tissues trough Runx1/Stat3 signalling network, while TF Ets2 together with p-STAT3 activation induce cathepsins K and B expression in human rheumatoid arthritis synovial fibroblasts (RASFs).
>
> + **(Q5.2)** "For the education dataset, can you explain why the pre-intervention selection bias is a potential issue?"
>
>   **A:** Thank you for this insightful question. For educational datasets, not using our method might lead to doubts about the causal effect of incentives on improving female students' performance. This is due to potential selection bias, as there exists a causal graph where the probability of receiving an intervention ($X_1$), the potential for improved performance ($X_2$), being female ($X_3$), and being an admitted female college student ($X_4$) are all interrelated, where graphically $X_1$, $X_2$, $X_3$ jointly point to $X_4$. Then,
> 	+ When we control for $X_3$ (i.e., control for $X_4$) during the analysis, it is akin to pre-selecting by gender before assigning different interventions. This may create spurious associations between $X_1$ and $X_2$.
> 	+ However, by using our method, we can clearly demonstrate that this association reflects the genuine stratified causal effect across genders, rather than being merely the result of selection bias.
> 	+ Therefore, our model provides a powerful tool for reliably distinguishing true causal relationships from selection bias (up to their equivalence class), enabling us to better address the challenges present in real-world data.
> 	+ To demonstrate the results more clearly, we have updated the data analysis results and added a table (Table 1) in the appendix explaining the corresponding variables.
>
>
> ---
> **(Q6)**  Minor comments about clarity.
>
> **A:**  Thank you for your careful reading. We have corrected all the noted typos. We have also carefully rewritten our technical sections in our updated manuscript.
>
> In specific response to your comments, **we introduce all technical concepts, such as maximal ancestral graphs (MAGs), before they are used.**
>
> We hope you find the updated manuscript easier to follow.

---

> ### Author Response · Authors · 2024-11-22
> **Response to Reviewer 5bYr (Part 4/4)**
>
> We are encouraged that the reviewer recognizes our work as tackling "an important yet often overlooked" problem, and that "the setting is general".
>
> We would like to highlight that our contribution goes beyond solving a specific problem; **we identify a new problem setting and establish a general framework to understand this setting, namely, interventional studies under selection bias.**
>
> In real-world scenarios, selection indeed always meets intervention, as real experiments are typically designed with specific scopes and objectives (e.g., participants in a drug trial are usually patients of the relevant disease).
>
> To this end, we hope this work provides **a foundation for various potential practical extensions**, such as incorporating latent variables for causal discovery and deriving causal inference results.
>
> **We want to thank the reviewer again for all the valuable feedback.**

---

> > ### Comment · Reviewer_5bYr · 2024-11-25
> >
> > Thank the authors for detailed responses, which have addressed most of my concerns. I have changed my score accordingly.

---

> ### Author Response · Authors · 2024-11-25
>
> We sincerely thank you for your constructive feedback, the time you have dedicated, and your recognition of our work. Thank you.

---

### Official Review · Reviewer_riv8 · 2024-11-04

**Soundness:** 3
**Presentation:** 3
**Contribution:** 3
**Rating:** 8
**Confidence:** 3

**Summary:**

This paper addresses the issue of selection bias in interventional causal discovery, highlighting how existing methods fall short when subjects are selectively enrolled in experiments. To address this, the authors introduce a new graphical representation called "interventional twin graph" that explicitly represents both the observed world (where interventions are applied) and the counterfactual world (where selection occurs). They characterize the Markov properties of the proposed graphical model and develop an algorithm built upon FCI based on the proposed graphical model and Markov properties, named CDIS (Causal Discovery from Interventional data under potential Selection bias). The authors prove the soundness of CDIS. Experiments conducted under selection bias conditions demonstrate that CDIS outperforms baselines on synthetic datasets and uncovers novel causal relationships in real-world applications.

**Strengths:**

1. The selection bias in interventional experiments is an under-explored but crucial issue in causal discovery, and the authors use two clear examples to illustrate why this problem matters and why the existing methods and simply augmenting DAG fail.

2. The authors provide a solid theoretical foundation in this paper by (1) rigorously defining the interventional twin graph and characterizing its Markov properties and (2) proving the soundness of the proposed algorithm.

3. Synthetic experiments show that the proposed method outperforms baselines in handling selection bias and remains robust as the number of variables increases. It also uncovers novel causal relationships in real-world applications.

**Weaknesses:**

1. While the introduction of the interventional twin graph and its Markov properties is rigorous, it may be challenging for readers to grasp at first glance due to its complexity. Providing a high-level explanation to offer an intuitive understanding would greatly benefit readers.

2. The interventional twin graph is more complex than a simple DAG and involves additional nodes. It would be helpful if the authors discussed the computational cost of the proposed model compared to the simpler DAG, including an analysis of the algorithm's computational complexity under the new graphical model.

3. The authors did not address the identifiability guarantees of the proposed method. It would be useful to know if the method can reliably identify the selection variables and under what conditions the true interventional twin graph can be identified.

4. Minor typos:
    * Line 48: "We show that existing existing graphical representation paradigms" --> "We show that existing graphical representation paradigms"
    * At the end of line 169: "models a completely different scenario" needs to be revised

**Questions:**

1. Please refer to the questions mentioned in Weaknesses.

2. I think theoretically the proposed framework (the proposed graphical model + algorithm) should be able to handle interventions without selection bias, but how does the proposed algorithm perform empirically compared to existing methods in scenarios without selection bias?

---

> ### Author Response · Authors · 2024-11-22
> **Response to Reviewer riv8 (Part 1/3)**
>
> We sincerely appreciate the reviewer's encouragement and insightful feedback. Please see below for our response.
>
> ---
> **(Q1)**  The reviewer suggests a high-level explanation for the Markov properties.
>
> **A:**  Thank you for your suggestion. In response, we have carefully rewritten our technical sections in the updated manuscript. We provide **each definition and theorem with graph examples and explanations afterwards**. All examples are based on the two running examples—the clinical and pest control ones—throughout, to illustrate the main results.
>
> For more details, please kindly refer to the updated manuscript.
>
> ---
> **(Q2)**  The reviewer wonders the computational cost of the algorithm.
>
> **A:**  Thank you for this insightful question. The computational cost of our algorithm, CDIS, is similar to standard constraint-based algorithms like PC and FCI, as it mainly involves conducting conditional independence (CI) tests **among observed variables to estimate adjacencies**. This has been mentioned in the updated manuscript. Specifically,
>  - **(Q2.1)** What is the exact time complexity?
>
>    **A:** Like PC, our algorithm has a worst-case complexity exponential in the number of observed variables (the $D$ in our manuscript). However, if the underlying graph is sparse—a common and reasonable assumption—the runtime becomes polynomial [1].
>
>  - **(Q2.2)** But the interventional twin graph is more complex than a simple DAG and involves additional nodes?
>
>    **A:** This is because the true data generating mechanism is indeed more complex than and cannot reduce to a simple augmented DAG, as shown in Section 2 (Motivations). Regarding complexity of the discovery procedure, note that the above mentioned complexity of the twin graph model only impacts **how we interpret** the CI relations from data, **not the number of CI tests required,** as we can only test observed variables.
>
>  - For empirical evaluation, we have also compared the runtime (in seconds) of our algorithm with others (even though they are theoretically incorrect under selection). Results are shown as follows:
>
> 	|      Method         | 10 nodes | 15 nodes | 20 nodes|
> 	|:--------------|:---------|:---------|:---------|
> 	| **CDIS**          | 0.4±0.24 | 1.0±0.32 | 2.0±0.0  |
> 	| IGSP          | 0.6±0.24 | 0.8±0.2  | 1.2±0.2  |
> 	| UT-IGSP       | 0.6±0.24 | 0.6±0.24 | 1.4±0.24 |
> 	| JCI-GSP       | 0.4±0.24 | 0.8±0.2  | 1.4±0.24 |
>
>      We observe that these algorithms indeed operate on a similar timescale.
>
> ---
> **(Q3)**  The reviewer has concerns about the identifiability guarantees of our algorithm.
>
> **A:**  We thank the reviewer for highlighting this, and would like to note that **we do have provided identifiability guarantees** in our work (Theorem 2 and Theorem 3), as detailed below:
>
>  - **Theorem 2** provides the **graphical criteria** for determining whether two DAGs are **Markov equivalent** under two collections of interventions, offering an upper bound on identifiability.
>  - **Theorem 3** demonstrates the **soundness of the proposed CDIS algorithm**, showing that it reliably identifies the true causal and selection relations. This guarantee is general, without requiring any additional structural or parametric assumptions.
>  - After Theorem 3, we also discuss the CDIS's completeness, i.e., whether it identifies all the invariant relations in the equivalence class. Brutal search strongly supports our conjecture of its completeness. However, proving this remains technically challenging, as outlined in the updated manuscript.
>
> We hope this addresses your concerns about identifiability guarantees. If not, please kindly let us know; we hope for the opportunity to discuss them further with you.

---

> ### Author Response · Authors · 2024-11-22
> **Response to Reviewer riv8 (Part 2/3)**
>
> **(Q4)**  The reviewer wonders how the proposed algorithm performs empirically compared to existing methods in scenarios without selection bias.
>
> **A:**  Thank you for this constructive comment. We conducted experiments on datasets without selection bias. **It is worth noting that we directly applied the original CDIS algorithm to the data, without making use of the prior knowledge about the absence of selection bias,** as otherwise the algorithm just reduces to other typical methods based on augmented graphs.
>
> This makes the choice of metrics particularly relevant. Recall that in our clinical example, a DAG $2\rightarrow 1$ without selection is unidentifiable to another DAG $1 \rightarrow S \leftarrow 2$ with selection, under the intervention on $X_1$. This shows that even without selection bias, when the prior knowledge of its absence is unknown, still not all edges in a DAG are identifiable.
>
> Hence, in what follows we **report results using both metrics** based on direct edges in the equivalence class (assuming no prior knowledge of selection absence), and all direct edges in the original DAG.
>
> The experiments were performed on varying numbers of nodes with 1,000 samples, with all other configurations the same as Section 5.1. We report the F1 score, precision, and recall under both metrics. The results are provided in the tables below and in Figure 13 in the updated manuscript.
>
>   + Using equivalence class metric:
> | #Nodes   |         10          |             |             |         15         |             |             |         20          |             |             |
> |----------|----------------------|-------------|-------------|--------------------|-------------|-------------|---------------------|-------------|-------------|
> |    **Method**      | **F1**               | **Precision** | **Recall**   | **F1**            | **Precision** | **Recall**   | **F1**              | **Precision** | **Recall**   |
> | **CDIS** | 0.73±0.09           | **0.73±0.08** | 0.74±0.11   | **0.73±0.04**     | **0.72±0.08** | 0.78±0.04   | **0.79±0.03**       | **0.72±0.05** | **0.88±0.02** |
> | IGSP     | 0.6±0.06            | 0.52±0.05    | 0.72±0.06   | 0.63±0.07         | 0.52±0.07    | **0.82±0.08** | 0.47±0.02          | 0.34±0.02    | 0.74±0.03   |
> | UT-IGSP  | **0.75±0.05**       | 0.68±0.06    | **0.84±0.04** | 0.61±0.06        | 0.51±0.06    | 0.79±0.07   | 0.51±0.04          | 0.37±0.03    | 0.79±0.05   |
> | JCI-GSP  | 0.39±0.06           | 0.3±0.05     | 0.58±0.05   | 0.39±0.04         | 0.27±0.04    | 0.69±0.03   | 0.25±0.03          | 0.16±0.02    | 0.55±0.03   |
>
>   + Using the original DAG metric:
> | #Nodes   |         10          |             |             |         15          |             |             |         20          |             |             |
> |----------|----------------------|-------------|-------------|--------------------|-------------|-------------|---------------------|-------------|-------------|
> |   **Method**       | **F1**               | **Precision** | **Recall**   | **F1**            | **Precision** | **Recall**   | **F1**              | **Precision** | **Recall**   |
> | **CDIS** | 0.73±0.09           | **0.73±0.08**    | 0.74±0.11   | **0.71±0.04**         | **0.73±0.08**    | 0.71±0.05   | **0.78±0.04**           | **0.76±0.04**    | **0.81±0.06**   |
> | IGSP     | 0.6±0.06            | 0.52±0.05    | 0.72±0.06   | 0.63±0.07         | 0.54±0.07    | 0.77±0.07   | 0.46±0.02           | 0.35±0.02    | 0.66±0.03   |
> | UT-IGSP  | **0.75±0.05**           | 0.68±0.06    | **0.84±0.04**   | 0.65±0.05         | 0.56±0.05    | **0.79±0.06**   | 0.53±0.05           | 0.41±0.04    | 0.75±0.04   |
> | JCI-GSP  | 0.39±0.06           | 0.3±0.05     | 0.58±0.05   | 0.41±0.04         | 0.3±0.04     | 0.68±0.04   | 0.27±0.02           | 0.18±0.02    | 0.55±0.02   |
>
> Directed edges in the equivalence class form a subset of those in the DAG. We anticipated that CDIS would perform similarly or better using the equivalence class metric but worse with the original DAG metric, as it lacks the prior assumptions of the absence of selection bias, while other methods do.
>
> Surprisingly, the results show that CDIS performs slightly better on both metrics. A possible explanation is that we simulate 10 interventional datasets, which provide sufficient information to identify more edges in the equivalence class. Motivation on this is shown in Example 4 in the updated manuscript.
>
>
> ---
> **(Q5)**  Minor typos.
>
> **A:**  Thank you for your careful reading. We have corrected all the noted typos in the updated manuscript.

---

> ### Author Response · Authors · 2024-11-22
> **Response to Reviewer riv8 (Part 3/3)**
>
> We are encouraged that the reviewer recognizes our work as tackling "an under-explored but crucial issue in causal discovery".
>
> We would like to highlight that our contribution goes beyond solving a specific problem; **we identify a new problem setting and establish a general framework to understand this setting, namely, interventional studies under selection bias.**
>
> In real-world scenarios, selection indeed always meets intervention, as real experiments are typically designed with specific scopes and objectives (e.g., participants in a drug trial are usually patients of the relevant disease).
>
> To this end, we hope this work provides **a foundation for various potential practical extensions**, such as incorporating latent variables for causal discovery and deriving causal inference results.
>
> **We want to thank the reviewer again for all the valuable feedback.**
>
>
> ---
> [1] Kalisch, Markus, and Peter Bühlman. "Estimating high-dimensional directed acyclic graphs with the PC-algorithm." (2007).

---

> ### Comment · Reviewer_riv8 · 2024-11-26
> **Official Comment by Reviewer riv8**
>
> Thank you for the detailed response and for providing the additional experimental results. All my concerns have been addressed, and I have updated my score accordingly.

---

> ### Author Response · Authors · 2024-11-27
>
> We sincerely appreciate your thoughtful feedback and recognition of our work. Thank you for your time.

---

### Official Review · Reviewer_VRoQ · 2024-11-05

**Soundness:** 3
**Presentation:** 3
**Contribution:** 4
**Rating:** 8
**Confidence:** 3

**Summary:**

The work shows that current theory for dealing with interventional causal discovery is insufficient under selection bias, as there are cases where the selection mechanism takes place before the intervention. The authors then propose a twin graph framework where the selection happens before the intervention and then define a Markov property on this twin graph that implies certain independences in distributions when conditioned on the selection mechanism. The authors then propose a method that constructs the graph based on the Markov property up to an equivalence class.

**Strengths:**

- The authors have found an interesting flaw in previous methods in the presence of selection bias
- The paper is well motivated with clear examples. Some examples can be made clearer (see below)

**Weaknesses:**

Some bits of the exposition are unclear (see below).

**Questions:**

- In example 2: This will be much more convincing if there is a numerical example that actually shows this (like example 1). Otherwise, it's not obvious to me why selection might imply that $X_1, X_3$ are dependent when conditioning on $X_2$.
- In section 3.2: It is claimed that interventional distributions are not Markovian due to the original dag due to $X_1$ not being independent of $X_3$ when $X_2$ is conditioned on. The open path here goes through $X_2^*$. However, in example 4 it is claimed that conditioning on $X_2$ automatically conditional on $X_2^*$, This seems contradictory to the previous statement, what am I missing here?
- Thm 1: If this is contrasted directly with the regular Markov property in DAGs, it will help clear up what the extra condition here is. Reading this section, the extra condition has not been motivated with respect to the examples shown. For example, how does this theorem differentiate DAGs $\mathcal{G}$ and $\mathcal{H}$.
- I don't really understand how Lemma 1 "misses key distributional information", could this be made more explicit?
- L291: Why does the twin graph have fewer degrees of freedom with the invariance constraint? And why does it lead to CIs not implied by d-separations? The claim in L293 is also not clear at all to me.
- Section 3.3: Intuitions behind the lemmas and why they are needed would greatly help here. Right now they are hardly motivated and tough to understand given the dense notation.
- Algorithm 1 step 2: how is this done? What method is used to test if the conditional distributions are the same or not?

---

> ### Author Response · Authors · 2024-11-12
> **[Fixed] Possible review submission error**
>
> Update: It has been fixed. Thank you.
>
> ---
>
> ~~Dear Reviewer VRoQ,~~
>
>
> ~~Thank you for your careful review. It appears the review intended for another submission may have been posted here by mistake. Could you please help us in resolving this?~~
>
>
> ~~Thank you,~~
>
> ~~-Authors of Submission 1361~~

---

> ### Author Response · Authors · 2024-11-22
> **Response to Reviewer VRoQ (Part 1/2)**
>
> We sincerely appreciate the reviewer's constructive comments and helpful feedback. Please see below for our response.
>
> ---
> **(Q1)**  The reviewer suggests a simulation to illustrate why the intervention destroys a conditional independence in Example 2 (pest control).
>
> **A:**  Thanks for the suggestion. We have **provided a simulation with scatterplots** for this example. Due to space limit, we put in Appendix B.1 in the updated manuscript. In brief, such loss of conditional independence is because that selection renders exogenous noise terms not independent anymore.
>
>
> ---
> **(Q2)**  The reviewer wonders how conditional distribution invariance is tested.
>
> **A:**  For two datasets $p^{(0)}$ and $p^{(1)}$, to test whether a conditional distribution is invariant, we introduce a binary dataset indicator variable $\zeta$, where $\zeta=0$ for samples from $p^{(0)}$, and $\zeta=1$ for samples from $p^{(1)}$. Let $p$ denote the stacked data of $p^{(0)}, p^{(1)}$. Then, the conditional distribution invariance $p^{(0)}(X_A|X_C) = p^{(1)}(X_A|X_C)$ is equivalent to the equality $p(X_A|X_C, \zeta=0) = p(X_A|X_C, \zeta=1)$, that is, the conditional independence relation $\zeta \unicode{x2AEB} X_A | X_C$ in stacked data $p$. The invariance of conditional distributions is thus tested by **testing for the corresponding conditional independence** involving $\zeta$ in the stacked data.
>
> As we have shown in Section 2.1, this is exactly the motivation to introduce $\zeta$ variables into the graph, so as to model conditional distribution invariance nonparametrically.
>
> In practical usage, while this can be realized by all kinds of conditional independence tests, we use Fisher's Z tests in our experimental runs for speed consideration.
>
> ---
> **(Q3)**  The reviewer suggests providing intuitions behind the lemmas.
>
> **A:**  Thank to your suggestions, we have carefully rewritten our technical sections in the updated manuscript. We introduce **each lemma alongside the specific question it aims to answer**, highlighted in blue in the revision. We hope you find the updated manuscript more readable.
>
> ---
> **(Q4)**  The reviewer asks for clarification on several technical points.
>
> **A:**  We appreciate your careful reading, and we understand how our previous presentation may have been hard to follow. In our original submission, we presented our theory development sequentially: starting with an intuitive model, then demonstrating its flaws, and finally making corrections to reach the optimal model. While this approach was helpful for understanding the technical progression, it may have impacted readability.
>
> Therefore, in the updated manuscript, we **directly present the optimal model**, which we believe resolves most of the confusion. Nonetheless, let us still address each of your points below. **Please note that the following responses are based on the original manuscript.**
>
>   - **(Q4.1)** Why was there a contradiction between Example 3 ("open path goes through $X_2^*$") and Example 4 ("...automatically conditions on $X_2^*$")?
>
>      **A:** This was because of the different intervention targets. In Example 3, the intervention targeted $X_1$, while in Example 4, it targeted $X_3$. This contrast was intended to highlight the insufficiency of relying solely on d-separations in the previous model: varying data behaviors under different interventions were described by the same graph.
>
>   - **(Q4.2)** What was the extra condition beyond regular Markov property in DAGs in Theorem 1?
>
>     **A:** The extra condition was that when an unaffected variable $X_i$ is conditioned on, its counterfactual part $X_i^*$ must also be conditioned, as unaffected variables retain their original values after intervention.
>
>   - **(Q4.3)** How did Lemma 1 miss key distributional information?
>
>     **A:** Lemma 1 showed that in the previous model, the $X^*$ variables could be marginalized out, resulting in selection directly applied to exogenous noise terms $\mathcal{E}$, leading to the loss on sparsity in the selection mechanisms. The motivation for this was given in Example 5, which showed two DAGs that are already distinguishable with only observational data. However, using the previous model, their corresponding interventional twin graphs entail the same sets of d-separations for arbitrary targets.
>
>
> Once again, we appreciate the reviewer's input and hope these explanations clarify our motivation behind the technical progression in our original submission. We also hope you find that **these potentially confusing points have already been avoided in the updated manuscript.**

---

> ### Author Response · Authors · 2024-11-22
> **Response to Reviewer VRoQ (Part 2/2)**
>
> We are encouraged that the reviewer recognizes our work as "finding an interesting flaw in previous methods".
>
> We would like to highlight that our contribution goes beyond solving a specific problem; **we identify a new problem setting and establish a general framework to understand this setting, namely, interventional studies under selection bias.**
>
> In real-world scenarios, selection indeed always meets intervention, as real experiments are typically designed with specific scopes and objectives (e.g., participants in a drug trial are usually patients of the relevant disease).
>
> To this end, we hope this work provides **a foundation for various potential practical extensions**, such as incorporating latent variables for causal discovery and deriving causal inference results.
>
> **We want to thank the reviewer again for all the valuable feedback.**

---

> > ### Comment · Reviewer_VRoQ · 2024-11-25
> > **Response to Author**
> >
> > I appreciate the changes that the authors have made. The example in Appendix B.1 is very helpful and section 3 does read a lot better.
> >
> > Nevertheless, I still find some of the writing verbose and certain sections are not well motivated. For example, in 2.3 states that the problem being studied is when selection is applied *before* data is observed or interventions are applied. Somehow, this is written much more verbosely ("interventions are never applied in a vacuum") that actually serves to obfuscate this idea.
> >
> > More motivation might help with these questions:
> > - Why are MAGs introduced? Twin graphs are used to explain the CI and invariance one might expect under selection, are these not already present in MAGs? This needs to be much more clearly explained.
> > - Is the "MAG of a twin graph", different to a regular MAG, if so, this needs to be explained. If it is different, would it make sense to name it something different?
> > - Is the definition of Markov equivalence of "MAG of a twin graph" different to the regular MAG? If not, could you explain the difference between your work and works that find the Markov equivalence of MAGs from data?
> >
> > I'm inclined to raise my score, as I think this is tackling an interesting and novel problem, if these issues above are addressed.

---

> ### Author Response · Authors · 2024-11-25
> **Follow-Up Response to Reviewer VRoQ (Part 1/2)**
>
> We sincerely appreciate the reviewer’s prompt and valuable feedback. Please find our detailed responses below:
>
> ---
>
> **(Q1)** The reviewer points out that "interventions are never applied in a vacuum" writing is unclear.
>
> **(A)** Thank you for this constructive comment. In light of it, we have revised the sentence in the updated manuscript to **"In real-world scenarios, interventions are usually applied *after* selection, as experiments are typically designed to target specific scopes of study."**
>
> Moreover, please let us share more motivation behind the original phrasing. Our intension was to differentiate the proposed twin graph model from the simpler extension of the augmented graph model. The latter assumes interventions are applied universally, without prior selection, and after that, different intervened individuals undergo a same selection mechanism.
>
> We aimed to highlight that this latter scenario is much less common in practice. Selection can often be understood as a survival process. For example, in biological experiments, only cells that meet specific criteria can survive, and **any cell has to first survive itself before receiving any interventions or observations.** In this sense, applying interventions without prior selection—what we referred to as "from a vacuum"—is rarely feasible in real-world settings.
>
> We hope this revision and clarification address the reviewer's concerns.
>
>
> ---
>
> **(Q2)** The reviewer wonders further motivation for introducing MAGs. Specifically,
>
> **(Q2.1)** Why are MAGs introduced? Aren't twin graphs sufficient to explain all the CIs and invariances?
>
> **(A)** Thank you for this thoughtful question. As is noted in our manuscript (line 312), **"we introduce MAGs to simplify the graphical representation for the CI implications."** Below, we provide a more detailed explanation, by showing the role that MAGs play throughout our manuscript.
>
> After establishing the interventional twin graph in Section 3.1, our manuscript is structured to address the following three key questions about it, step by step:
>
>    1. What are the CI implications of an interventional twin graph? (Section 3.2)
>    2. When do two interventional twin graphs entail the same CI implications? (Section 3.3)
>    3. How, and to what extent, can we infer the true interventional twin graph from data? (Section 4)
>
> **To understand why we introduce MAGs, let us examine the role that MAGs play in addressing each of these questions.**
>
>    1. For question 1, **MAGs provide a more intuitive graphical representation to interpret CI implications.** While it is true that d-separations in interventional twin graphs (which are DAGs) can already characterize all CIs and invariances, they can be unintuitive to interpret. To see this, recall that in a fully observed DAG, two variables are adjacent if and only if they are dependent given any other variable sets. However, this no longer holds with latent and selection variables. For instance, in the pest control example, $X_1$ and $X_3$ are not adjacent in the twin graph but remain conditionally dependent in all cases. MAGs address, where two variables are adjacent if and only if they are always dependent, simplifying interpretation (Lemma 4 in the updated manuscript).
>    2. For question 2, **MAGs allow Markov equivalence to be determined through clear and efficient graphical criteria**, such as adjacencies and v-structures, as presented in Theorem 2. By analogy, for two fully observed DAGs, equivalence can be defined in two graphical criteria: (1) "they have the same d-separations", or (2) "they have the same adjacencies and v-structures". Though the two are equivalent, the latter is obviously more intuitive and efficient. Similarly, without MAGs, equivalence in interventional twin graphs could only be defined through d-separations on DAGs with latent and selection variables, which can become unnecessarily complex.
>    3. For question 3, **MAGs serve as a graphical object where algorithms can be grounded.** Without MAGs, inferring the structure of a DAG from partially observed data becomes significantly challenging, as relationships involving latent and selection variables can be arbitrarily complex while they are completely inaccessible. MAGs, as graphical representations over observed variables only, provide a grounded structure that allows CI relations to be effectively used for causal discovery. The process is made even more practical with well-established algorithms like FCI based on MAGs.

---

> ### Author Response · Authors · 2024-11-25
> **Follow-Up Response to Reviewer VRoQ (Part 2/2)**
>
> **(Q2.2)** Is the "MAG of a twin graph" different to a regular MAG?
>
> **(A)** **The "MAG of a twin graph" is not fundamentally different from a regular MAG. It is simply a specific class of regular MAGs.** By analogy, an interventional twin graph is a regular DAG—meaning all its graphical properties like d-separations are the same as a regular DAG—while it is a specific class of DAGs with its form constrained in Definition 1.
>
> The reason why we specifically define the "MAG of a twin graph" is **to better understand the CI implications graphically, and to eventually identify more information from data**, using knowledge about the structural constraints on interventional twin graphs.
>
> For example, to answer the question, "when does an intervention alter all of a variable’s conditional distributions?" the general MAG adjacency rule (Definition 3, though being correct, can be too broad, whereas Lemma 7 directly answers that "this occurs if and only if the variable is directly targeted, or both affected and ancestrally selected." Moreover, with the specific orientation rules (Lemma 6), we can eventually identify more information (in terms of edge orientations) than a standard FCI can from data.
>
>
> ---
>
> **(Q2.3)** Is the definition of Markov equivalence of "MAG of a twin graph" different to the regular MAG?
>
> **(A)** Thank you for this especially insightful question. **Yes, they are different.** As shown in our Definition 2, two MAGs of interventional twin graphs are equivalent, if and only if they have the same adjacencies and v-structures. In contrast, for two MAGs of arbitrary DAGs, these two conditions are necessary but not sufficient; an additional condition is required. That additional condition is relatively complicated, so please bear with us to just quote it here without explanation: "If a path $p$ is a discriminating path for a vertex $V$ in both MAGs, then $V$ must be a collider on the path in one MAG if and only if it is a collider on the path in the other."
>
> The reason why we can avoid this third condition is, again, due to the specific structural constraints of interventional twin graphs, from which we prove that this third condition is always satisfied for MAGs of interventional twin graphs.
>
>
> All the three questions in **(Q2)** have been incorporated into our updated manuscript (lines 312, 352, and Footnote 4, respectively). We hope these revisions and clarifications address the reviewer's concerns. If not, we always look forward to the opportunity to discuss them further with you.
>
>
> ---
>
> Once again, we want to thank the reviewer for your thoughtful questions, and your continuous input that helps improving the presentation of our submission.

---

> > ### Comment · Reviewer_VRoQ · 2024-11-25
> > **Response to Authors (2)**
> >
> > Thank you for the clarification. Might it make sense to name "MAG of a twin graph" something? A reader may be confused into thinking that that the experimental section of this paper is discovering MAGs, whereas you are discovering (the Markov equivalence of) a specific type of MAG with specific properties (derived from the twin interventional graph). Ignore me if this interpretation is incorrect.
> >
> > I thank the authors for their thorough responses and have raised my score accordingly.

---

> ### Author Response · Authors · 2024-11-25
>
> We sincerely thank you for your recognition of our work. Your summary of our algorithm is correct and sharp!
>
> In light of your suggestion, we are considering giving two separate definitions as "interventional twin DAGs" and "interventional twin MAGs." The latter, however, may not be entirely appropriate, as the MAG includes only observed variables and does not contain the twin world. We are still looking for a better way to present this. Once finalized, we will let you know and update the manuscript accordingly.
>
> Thank you again for this constructive suggestion.

---

### Meta-Review · Area_Chair_wM8f · 2024-12-21

**Metareview:**

The paper identifies limitations of existing causal discovery methods under pre-intervention selection bias and proposes the interventional twin graph framework to explicitly model both observed and counterfactual worlds, defining Markov properties and sound algorithms for causal inference.

Strengths:

+ Tackles the critical but underexplored issue of selection bias in interventional causal discovery.

+ Defines the twin graph framework and establishes its Markov properties, with proofs for soundness.


Weaknesses:

+ The dense notation and complexity of the twin graph model make it challenging for readers to grasp intuitively.

+ Insufficient analysis of computational costs, algorithmic completeness, and identifiability guarantees.

+ Lack of strong reporting and comparison of empirical results; completeness and sensitivity analysis, are needed to strengthen the conclusions.

**Additional Comments On Reviewer Discussion:**

The reviewers unanimously agree that it is a good paper.

---

### Decision · Program_Chairs · 2025-01-22

Accept (Oral)